# PRESTOPPING: HOW DOES EARLY STOPPING HELP GENERALIZATION AGAINST LABEL NOISE?

## ABSTRACT

Noisy labels are very common in real-world training data, which lead to poor generalization on test data because of overfitting to the noisy labels. In this paper, we claim that such overfitting can be avoided by "early stopping" training a deep neural network before the noisy labels are severely memorized. Then, we resume training the early stopped network using a "maximal safe set," which maintains a collection of almost certainly true-labeled samples at each epoch since the early stop point. Putting them all together, our novel two-phase training method, called **_Prestopping_**, realizes _noise-free_ training under _any type_ of label noise for practical use. Extensive experiments using four image benchmark data sets verify that our method significantly outperforms four state-of-the-art methods in test error by 0.4–8.2 percent points under existence of real-world noise.

## 1 INTRODUCTION

By virtue of massive labeled data, deep neural networks (DNNs) have achieved a remarkable success in numerous machine learning tasks, such as image classification (Krizhevsky et al., 2012) and object detection (Redmon et al., 2016). However, owing to their high capacity to memorize any label noise, the generalization performance of DNNs drastically falls down when noisy labels are contained in the training data (Jiang et al., 2018; Han et al., 2018; Song et al., 2019). In particular, Zhang et al. (2017) have shown that a standard convolutional neural network (CNN) can easily fit the entire training data with any ratio of noisy labels and eventually leads to very poor generalization on the test data. Thus, it is challenging to train a DNN robustly even when noisy labels exist in the training data.

A popular approach to dealing with noisy labels is "sample selection" that selects true-labeled samples from the noisy training data (Jiang et al., 2018; Ren et al., 2018; Han et al., 2018; Yu et al., 2019; Song et al., 2019). Here, $(1-\tau)\times100\%$ of _small-loss_ training samples are treated as true-labeled ones and then used to update a DNN robustly, where $\tau \in [0,1]$ is a noise rate. This _loss-based separation_ is well known to be justified by the _memorization effect_ (Arpit et al., 2017) that DNNs tend to learn easy patterns first and then gradually memorize all samples. In practice, Han et al. (2018) empirically proved that training on such small-loss samples yields a much better generalization performance under artificial noise scenarios.

Despite its great success, Song et al. (2019) have recently argued that the performance of the loss-based separation becomes considerably worse depending on the type of label noise. For instance,

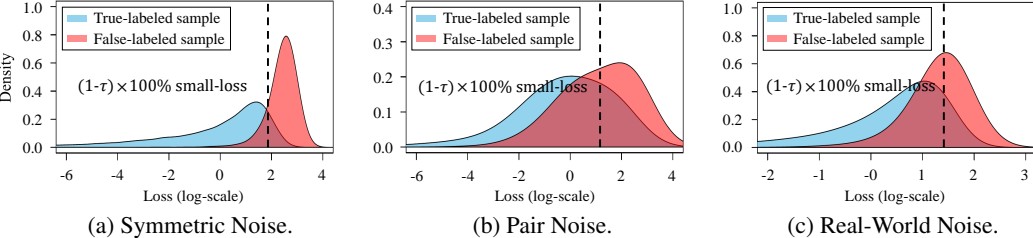

(a) Symmetric Noise.      (b) Pair Noise.      (c) Real-World Noise.

Figure 1: Loss distributions at the training accuracy of $50\%$ using DenseNet (L=40, k=12): (a) and (b) show those on CIFAR-100 with two types of synthetic noises of $40\%$, where "symmetric noise" flips a true label into other labels with equal probability, and "pair noise" flips a true label into a specific false label; (c) shows those on FOOD-101N (Lee et al., 2018) with real-world noise of $18.4\%$.

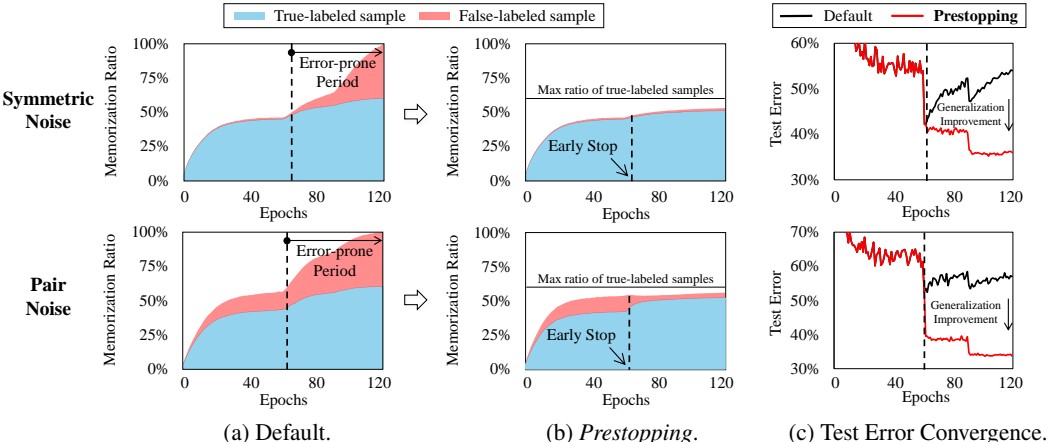

Figure 2: Key idea of *Prestopping*: (a) and (b) show how many true-labeled and false-labeled samples are memorized when training DenseNet (L=40, k=12)[1] on CIFAR-100 with two types of synthetic noises of $40\%$. "Default" is a standard training method, and "*Prestopping*" is our proposed one; (c) contrasts the convergence of test error between the two methods.

in *symmetric noise* (Figure 1(a)), the loss-based approach well separates true-labeled samples from false-labeled ones because many true-labeled ones exhibit smaller loss than false-labeled ones. On the other hand, in *pair* and *real-world noises* (Figures 1(b) and 1(c)), many false-labeled samples are misclassified as true-labeled ones because the two distributions overlap closely; this overlap confirms that the loss-based separation still accumulates severe label noise from many misclassified cases, especially in real-world noise or pair noise which is regarded as more *realistic* than symmetric noise (Ren et al., 2018; Yu et al., 2019). This limitation definitely calls for a new approach that supports *any type* of label noise for practical use.

In this regard, as shown in Figure 2(a), we thoroughly investigated the memorization effect of a DNN on the two types of noises and found two interesting properties as follows:

- **A noise type affects the memorization rate for false-labeled samples:** The memorization rate for false-labeled samples is faster with pair noise than with symmetric noise. That is, the red portion in Figure 2(a) starts to appear earlier in pair noise than in symmetric noise. This observation supports the significant overlap of true-labeled and false-labeled samples in Figure 1(b). Thus, the loss-based separation performs well only if the false-labeled samples are scarcely learned at an early stage of training, as in symmetric noise.

- **There is a period where the network accumulates the label noise severely:** Regardless of the noise type, the memorization of false-labeled samples significantly increases at a late stage of training. That is, the red portion in Figure 2(a) increases rapidly after the dashed line, in which we call the *error-prone period*. (See Section 3.2.1 for the details of estimating the error-prone period). We note that the training in that period brings *no benefit*. The generalization performance of "Default" deteriorates sharply, as shown in Figure 2(c).

Based on these findings, we contend that eliminating this error-prone period should make a profound impact on robust optimization.

In this paper, we propose a novel approach, called **Prestopping**, that achieves *noise-free* training based on the *early stopping* mechanism. Because there is no benefit from the error-prone period, *Prestopping* early stops training before that period begins. This early stopping effectively prevents a network from overfitting to false-labeled samples, and the samples memorized until that point are added to a *maximal safe set* because they are true-labeled (i.e., blue in Figure 2(a)) with high precision. Then, *Prestopping* resumes training the early stopped network *only* using the maximal safe set in support of noise-free training. Notably, our proposed merger of "early stopping" and "learning from the maximal safe set" indeed eliminates the error-prone period from the training process, as shown in Figure 2(b). As a result, the generalization performance of a DNN remarkably improves in both noise types, as shown in Figure 2(c).

---

[1]The learning rate, as usual, was decayed at $50\%$ and $75\%$ of the total number of training epochs.

To validate the superiority of *Prestopping*, DenseNet (Huang et al., 2017) and VGG-19 (Simonyan & Zisserman, 2015) were trained on both simulated and real-world noisy data sets, including CIFAR-10, CIFAR-100, ANIMAL-10N, and Food-101N. Compared with four state-of-the-art methods, *Prestopping* significantly improved test error by up to $18.1pp^2$ in a wide range of noise rates.

## 2  RELATED WORK

Numerous studies have been conducted to address the problem of learning from noisy labels. A typical method is using "loss correction" that estimates the label transition matrix and corrects the loss of the samples in the mini-batch. *Bootstrap* (Reed et al., 2015) updates the network based on their own reconstruction-based objective with the notion of perceptual consistency. *F-correction* (Patrini et al., 2017) reweights the forward or backward loss of the training samples based on the label transition matrix estimated by a pre-trained normal network. [R3:] *D2L* (Ma et al., 2018) employs a simple measure called local intrinsic dimensionality and then uses it to modify the forward loss in order to reduce the effects of false-labeled samples in learning. *Active Bias* (Chang et al., 2017) heuristically evaluates uncertain samples with high prediction variances and then gives higher weights to their backward losses. Ren et al. (2018) include small clean validation data into the training data and re-weight the backward loss of the mini-batch samples such that the updated gradient minimizes the loss of those validation data. However, this family of methods is known to accumulate severe noise from the *false correction*, especially when the number of classes or the number of false-labeled samples is large (Han et al., 2018; Yu et al., 2019).

To be free from the false correction, many recent researches have adopted "sample selection" that trains the network on selected samples. These methods attempt to select true-labeled samples from the noisy training data for use in updating the network. *Decouple* (Malach & Shalev-Shwartz, 2017) maintains two networks simultaneously and updates the models only using the samples that have different label predictions from these two networks. [R3:] Wang et al. (2018) proposed an iterative learning framework that learns deep discriminative features from well-classified noisy samples based on the local outlier factor algorithm (Breunig et al., 2000). *MentorNet* (Jiang et al., 2018) introduces a collaborative learning paradigm that a pre-trained mentor network guides the training of a student network. Based on the small-loss criteria, the mentor network provides the student network with the samples whose label is probably correct. *Co-teaching* (Han et al., 2018) and *Co-teaching+* (Yu et al., 2019) also maintain two networks, but each network selects a certain number of small-loss samples and feeds them to its peer network for further training. Compared with *Co-teaching*, *Co-teaching+* further employs the disagreement strategy of *Decouple*. [R1:] *ITLM* (Shen & Sanghavi, 2019) iteratively minimizes the trimmed loss by alternating between selecting a fraction of small-loss samples at current moment and retraining the network using them. *INCV* (Chen et al., 2019) randomly divides the noisy training data and then utilizes cross-validation to classify true-labeled samples with removing large-loss samples at each iteration. However, their general philosophy of selecting *small-loss* samples works well only in some cases such as symmetric noise. [RA:] Differently to this family, we exploit the maximal safe set initially derived from the memorized samples at the early stop point.

Most recently, a hybrid of "loss correction" and "sample selection" approaches was proposed by Song et al. (2019). Their algorithm called *SELFIE* trains the network on selectively refurbished false-labeled samples together with small-loss samples. *SELFIE* not only minimizes the number of falsely corrected samples but also exploits full exploration of the entire training data. Its component for sample selection can be replaced by our method to further improve the performance.

[RA:] For the completeness of the survey, we mention the work that provides an empirical or a theoretical analysis on why early stopping helps learn with the label noise. Oymak et al. (2019) and Hendrycks et al. (2019) have argued that early stopping is a suitable strategy because the network eventually begins to memorize all noisy samples if it is trained too long. Li et al. (2019) have theoretically proved that the network memorizes false-labeled samples at a later stage of training, and thus claimed that the early stopped network is fairly robust to the label noise. However, they did not mention how to take advantage of the early stopped network for further training. Please note that *Prestopping* adopts early stopping to derive a *seed* for the maximal safe set, which is exploited to achieve noise-free training during the remaining learning period. Thus, our novelty lies in the "merger" of early stopping and learning from the maximal safe set.

---

[2] A *pp* is the abbreviation of a percentage point.

## 3 ROBUST TRAINING VIA *Prestopping*

### 3.1 PRELIMINARIES

A $k$-class classification problem requires the training data $\mathcal{D} = \{x_i, y_i^*\}_{i=1}^N$, where $x_i$ is a sample and $y_i^* \in \{1, 2, \ldots, k\}$ is its *true* label. Following the label noise scenario, let's consider the noisy training data $\tilde{\mathcal{D}} = \{x_i, \tilde{y}_i\}_{i=1}^N$, where $\tilde{y}_i \in \{1, 2, \ldots, k\}$ is a *noisy* label which may not be true. Then, in conventional training, when a mini-batch $\mathcal{B}_t = \{x_i, \tilde{y}_i\}_{i=1}^b$ consists of $b$ samples randomly drawn from the noisy training data $\tilde{\mathcal{D}}$ at time $t$, the network parameter $\theta_t$ is updated in the descent direction of the expected loss on the mini-batch $\mathcal{B}_t$ as in Eq. (1), where $\alpha$ is a learning rate and $\mathcal{L}$ is a loss function.

$$\theta_{t+1} = \theta_t - \alpha \nabla \big( \frac{1}{|\mathcal{B}_t|} \sum_{x \in \mathcal{B}_t} \mathcal{L}(x, \tilde{y}; \theta_t) \big) \tag{1}$$

As for the notion of network memorization, a sample $x$ is defined to be *memorized* by a network if the majority of its recent predictions at time $t$ coincide with the given label, as in Definition 3.1.

**Definition 3.1. (Memorized Sample)** Let $\hat{y}_t = \Phi(x|\theta_t)$ be the predicted label of a sample $x$ at time $t$ and $H_x^t(q) = \{\hat{y}_{t_1}, \hat{y}_{t_2}, \ldots, \hat{y}_{t_q}\}$ be the history of the sample $x$ that stores the predicted labels of the recent $q$ times, where $\Phi$ is a neural network. Next, $P(y|x, t; q)$ is formulated such that it provides the probability of the label $y \in \{1, 2, ..., k\}$ estimated as the label of the sample $x$ based on $H_x^t$ as in Eq. (2), where $[\cdot]$ is the Iverson bracket[3].

$$P(y|x, t; q) = \frac{\sum_{\hat{y} \in H_x^t(q)} [\hat{y} = y]}{|H_x^t(q)|} \tag{2}$$

Then, the sample $x$ with its noisy label $\tilde{y}$ is a *memorized sample* of the network with the parameter $\theta_t$ at time $t$ if the condition in Eq. (3) holds.

$$\mathrm{argmax}_y P(y|x, t; q) = \tilde{y} \quad \square \tag{3}$$

### 3.2 TWO MAIN COMPONENTS AND PHASES

The key idea of *Prestopping* is learning from a maximal safe set with an early stopped network. Thus, the two components of "early stopping" and "learning from the maximal safe set" respectively raise the questions about **(Q1)** when is the best point to early stop the training process? and **(Q2)** what is the maximal safe set to enable noise-free training during the remaining period?

#### 3.2.1 QUESTION 1: BEST POINT TO EARLY STOP

It is desirable to stop the training process at the point when the network *(i)* not only accumulates *little* noise from the false-labeled samples, *(ii)* but also acquires *sufficient* information from the true-labeled ones. Intuitively speaking, as indicated by the dashed line in Figure 3(a), the best stop point is the moment when *label precision* and *label recall* in Definition 3.2 cross with each other because it is the best trade-off between the two metrics. The period beyond this point is what we call the *error-prone period* because label precision starts decreasing rapidly.

If the ground truth $y^*$ of a noisy label $\tilde{y}$ is known, the best stop point can be easily calculated by these metrics.

**Definition 3.2. (Label Metrics)** Let $\mathcal{M}_t \subseteq \tilde{\mathcal{D}}$ be a set of memorized samples at time $t$. Then, *label precision (LP)* and *recall (LR)* (Han et al., 2018) are formulated as Eq. (4).

$$\mathrm{LP} = \frac{|\{(x, \tilde{y}) \in \mathcal{M}_t : \tilde{y} = y^*\}|}{|\mathcal{M}_t|}, \quad \mathrm{LR} = \frac{|\{(x, \tilde{y}) \in \mathcal{M}_t : \tilde{y} = y^*\}|}{|\{(x, \tilde{y}) \in \tilde{\mathcal{D}} : \tilde{y} = y^*\}|} \quad \square \tag{4}$$

However, it is not straightforward to find the exact best stop point *without* the ground-truth labels. Hence, we present two practical heuristics of approximating the best stop point with *minimal supervision*. These two heuristics require either a small clean validation set or a noise rate $\tau$ for the minimal supervision, where they are widely regarded as available in many studies (Veit et al., 2017; Ren et al., 2018; Han et al., 2018; Yu et al., 2019; Song et al., 2019).

---

[3]The Iverson bracket $[P]$ returns 1 if $P$ is true; 0 otherwise.

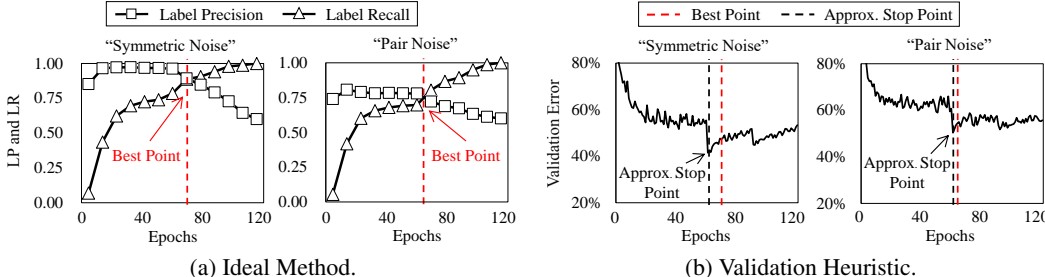

Figure 3: Early stop point estimated by ideal and heuristic methods when training DenseNet (L=40, k=12) on CIFAR-100 with two types of synthetic noises of $40\%$: (a) and (b) show the stop point derived by the ground-truth labels and the clean validation set, respectively.

- **Validation Heuristic:** If a clean validation set is given, we stop training the network when the validation error is the *lowest*. It is reasonable to expect that the lowest validation error is achieved near the cross point; after that point (i.e., in the error-prone period), the validation error likely increases because the network will be overfitted to the false-labeled samples. As shown in Figure 3(b), the estimated stop point is fairly close to the best stop point in both noise types.

- **Noise-Rate Heuristic:** If a noise rate $\tau$ is known, we stop training the network when the training error reaches $\tau \times 100\%$. If we assume that all true-labeled samples of $(1-\tau) \times 100\%$ are memorized before any false-labeled samples of $\tau \times 100\%$ (Arpit et al., 2017), this point indicates the cross point of label precision and label recall with their values to be all $1$. This heuristic tends to perform worse than the validation heuristic because the assumption does not hold perfectly. We further discuss its quality in Appendix A.

### 3.2.2 QUESTION 2: CRITERION OF A MAXIMAL SAFE SET

Because the network is early stopped at the (estimated) best point, the set of memorized samples at that time is quantitatively sufficient and qualitatively less noisy. That is, it can be used as a safe and effective training set to resume the training of the early stopped network without accumulating the label noise. Based on this intuition, we define a *maximal safe set* in Definition 3.3, which is initially derived from the memorized samples at the early stop point and gradually increased as well as purified along with the network's learning progress. In each mini-batch $\mathcal{B}_t$, the network parameter $\theta_t$ is updated using the current maximal safe set $\mathcal{S}_t$ as in Eq. (5), and subsequently a more refined maximal safe set $\mathcal{S}_{t+1}$ is derived by the updated network.

**Definition 3.3.** **(Maximal Safe Set)** Let $t_{stop}$ be the early stop point. A *maximal safe set* $\mathcal{S}_t$ at time $t$ is defined to be the set of the memorized samples of the network $\Phi(x; \theta_t)$ when $t \geq t_{stop}$. The network $\Phi(x; \theta_t)$ at $t = t_{stop}$ is the early stopped network, i.e., $\mathcal{S}_{t_{stop}} = \mathcal{M}_{t_{stop}}$, and the network $\Phi(x; \theta_t)$ at $t > t_{stop}$ is obtained by Eq. (5).

$$\theta_{t+1} = \theta_t - \alpha \nabla \big( \frac{1}{|\mathcal{B}'_t|} \sum_{x \in \mathcal{B}'_t} \mathcal{L}(x, \tilde{y}; \theta_t) \big)$$
$$\mathcal{B}'_t = \{x | x \in \mathcal{S}_t \cap \mathcal{B}_t\} \quad \square$$

(5)

Figure 4 shows the main advantage of learning from the maximal safe set during the remaining period. Even when the noise rate is quite high (e.g., $40\%$), this learning paradigm exploits most of true-labeled samples, in considering that the label recall of the maximal safe set was maintained very high in both CIFAR data sets regardless of the noise type. The label recall was maintained over $0.81$ in symmetric noise and over $0.84$ in pair noise even in CIFAR-100 with pair noise after the 80th epoch. Further, by excluding unsafe samples that might accumulate the label noise, the label precision of the maximal safe set tended to increase rapidly at the beginning of the remaining period; it was maintained over $0.98$ in symmetric noise and over $0.94$ even in CIFAR-100 with pair noise after the 80th epoch, which could *not* be realized by the small-loss trick (Han et al., 2018; Song et al., 2019; Shen & Sanghavi, 2019; Chen et al., 2019). [R2:]In Section 5.3, we have empirically verified that the maximal safe set is superior to the selected samples based on the small-loss trick.

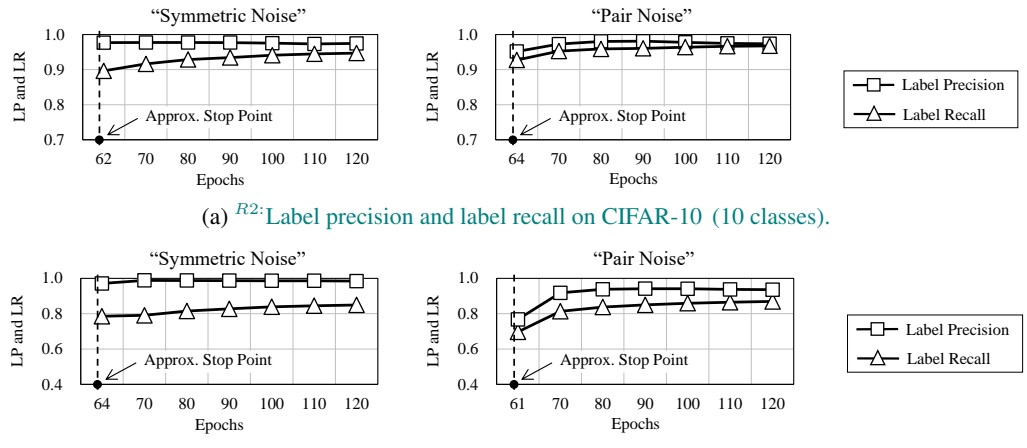

(a) [R2:]Label precision and label recall on CIFAR-10 (10 classes).

(b) Label precision and label recall on CIFAR-100 (100 classes).

Figure 4: Label precision and label recall of the maximal safe set during the remaining epochs when training DenseNet (L=40, k=12) on CIFAR data sets with two types of synthetic noises of $40\%$.

---

**Algorithm 1** *Prestopping* with **Validation** Heuristic

---

INPUT: $\tilde{\mathcal{D}}$: data, $\mathcal{V}$: clean validation data, $epochs$: total number of epochs, $q$: history length
OUTPUT: $\theta_t$: network parameters
1: $t \leftarrow 1$; $\theta_t \leftarrow$ Initialize the network parameter;
2: $\theta_{t_{stop}} \leftarrow \emptyset$; /* The parameter of the stopped network */
3: **for** $i = 1$ **to** $epochs$ **do** /* **Phase I: Learning from a noisy training data set** */
4:      **for** $j = 1$ **to** $|\tilde{\mathcal{D}}|/|\mathcal{B}_t|$ **do**
5:          Draw a mini-batch $\mathcal{B}_t$ from $\tilde{\mathcal{D}}$;
6:          $\theta_{t+1} = \theta_t - \alpha\nabla\big(\frac{1}{|\mathcal{B}_t|}\sum_{x\in\mathcal{B}_t}\mathcal{L}(x,\tilde{y};\theta_t)\big)$; /* Update by Eq. (1) */
7:          $val\_err \leftarrow$ Get_Validation_Error($\mathcal{V}, \theta_t$); /* A validation error at time $t$ */
8:          **if** $isMin(val\_err)$ **then** $\theta_{t_{stop}} \leftarrow \theta_t$; /*Save the network when $val\_error$ is the lowest*/
9:          $t \leftarrow t + 1$;
10: $\theta_t \leftarrow \theta_{t_{stop}}$; /* Load the network stopped at $t_{stop}$ */
11: **for** $i = stop\_epoch$ **to** $epochs$ **do** /* **Phase II: Learning from a maximal safe set** */
12:      **for** $j = 1$ **to** $|\tilde{\mathcal{D}}|/|\mathcal{B}_t|$ **do**
13:          Draw a mini-batch $\mathcal{B}_t$ from $\tilde{\mathcal{D}}$;
14:          $\mathcal{S}_t \leftarrow \{x|\text{argmax}_y P(y|x,t;q) = \tilde{y}\}$; /* A maximal safe set in Definition 3.3 */
15:          $\theta_{t+1} = \theta_t - \alpha\nabla\big(\frac{1}{|\mathcal{S}_t\cap\mathcal{B}_t|}\sum_{x\in\mathcal{S}_t\cap\mathcal{B}_t}\mathcal{L}(x,\tilde{y};\theta_t)\big)$; /* Update by Eq. (5) */
16:          $t \leftarrow t + 1$;
17: **return** $\theta_t, \mathcal{S}_t$;

---

## 4 ALGORITHM DESCRIPTION

### 4.1 MAIN ALGORITHM: *Prestopping* WITH VALIDATION HEURISTIC

Algorithm 1 describes the overall procedure of *Prestopping* with the *validation* heuristic, which is self-explanatory. First, the network is trained on the noisy training data $\tilde{\mathcal{D}}$ in the *default* manner (Lines 3–6). During this first phase, the validation data $\mathcal{V}$ is used to evaluate the best point for the early stop, and the network parameter is saved at the time of the lowest validation error (Lines 7–8). Then, during the second phase, *Prestopping* continues to train the early stopped network for the remaining learning period (Lines 10–12). Here, the maximal safe set $\mathcal{S}_t$ at the current moment is retrieved, and each sample $x \in \mathcal{S}_t \cap \mathcal{B}_t$ is used to update the network parameter. The mini-batch samples not included in $\mathcal{S}_t$ are no longer used in pursuit of robust learning (Lines 14–15).

### 4.2 COLLABORATION WITH SAMPLE REFURBISHMENT: *Prestopping+*

To further improve the performance of *Prestopping*, we introduce *Prestopping+* that employs the concept of selectively refurbishing false-labeled samples in *SELFIE* (Song et al., 2019). In detail,

the final maximal safe set $\mathcal{S}_{t_{end}}$ is retrieved from the first run of *Prestopping*, and then it is used as the true-labeled set for the next run of *SELFIE* with initializing the network parameter. Following the update principle of *SELFIE*, the modified gradient update rule in Eq. (6) is used to train the network. For each sample $x$ in the mini-batch $\mathcal{B}_t$, the given label $\tilde{y}$ of $x$ is selectively refurbished into $y^{refurb}$ if it can be corrected with high precision. However, the mini-batch samples included in $\mathcal{S}_{t_{end}}$ are omitted from the refurbishment because their label is already highly credible. Then, the mini-batch samples in the refurbished set $\mathcal{R}_t$ are provided to update the network together with those in the final maximal safe set $\mathcal{S}_{t_{end}}$. Refer to Song et al. (2019)'s work for more details.

$$\theta_{t+1} = \theta_t - \alpha \nabla \Big( \frac{1}{|\{x \in \mathcal{B}_t | x \in \mathcal{R}_t \cup \mathcal{S}_{t_{end}}\}|} \Big( \sum_{x \in \mathcal{R}_t \cup \mathcal{S}_{t_{end}}^c} \mathcal{L}(x, y^{refurb}; \theta_t) + \sum_{x \in \mathcal{S}_{t_{end}}} \mathcal{L}(x, \tilde{y}; \theta_t)) \Big) \quad (6)$$

## 5 EVALUATION

**Data Sets**: To verify the superiority of *Prestopping*, we performed an image classification task on *four* benchmark data sets: CIFAR-10 (10 classes)[4] and CIFAR-100 (100 classes)[4], a subset of 80 million categorical images (Krizhevsky et al., 2014); ANIMAL-10N (10 classes)[5], a real-world noisy data of human-labeled online images for confusing animals (Song et al., 2019); FOOD-101N (101 classes)[6], a real-world noisy data of crawled food images annotated by their search keywords in the FOOD-101 taxonomy (Bossard et al., 2014; Lee et al., 2018). We did not apply any data augmentation.

**Noise Injection**: As all labels in the CIFAR data sets are clean, we artificially corrupted the labels in these data sets using typical methods for the evaluation of synthetic noises (Ren et al., 2018; Han et al., 2018; Yu et al., 2019). For $k$ classes, we applied the label transition matrix $\mathbf{T}$: *(i) symmetric noise*: $\forall_{j \neq i} \mathbf{T}_{ij} = \frac{\tau}{k-1}$ and *(ii) pair noise*: $\exists_{j \neq i} \mathbf{T}_{ij} = \tau \wedge \forall_{k \neq i, k \neq j} \mathbf{T}_{ik} = 0$, where $\mathbf{T}_{ij}$ is the probability of the true label $i$ being flipped to the corrupted label $j$ and $\tau$ is the noise rate. For the pair noise, the corrupted label $j$ was set to be the next label of the true label $i$ following the recent work (Yu et al., 2019; Song et al., 2019). To evaluate the robustness on varying noise rates from light noise to heavy noise, we tested five noise rates $\tau \in \{0.0, 0.1, 0.2, 0.3, 0.4\}$. In contrast, we did not inject any label noise into ANIMAL-10N and FOOD-101N[7] because they contain real label noise estimated at $8.0\%$ and $18.4\%$ respectively (Lee et al., 2018; Song et al., 2019).

**Clean Validation Data**: Recall that a clean validation set is needed for the validation heuristic in Section 3.2.1. As for ANIMAL-10N and Food-101N, we did exploit their own clean validation data; $5,000$ and $3,824$ images, respectively, were included in their validation set. However, as no validation data exists for the CIFAR data sets, we constructed a small clean validation set by randomly selecting $1,000$ images from the *original* training data of $50,000$ images. Please note that the noise injection process was applied to only the rest $49,000$ training images.

**Networks and Hyperparameters**: For the classification task, we trained DenseNet (L=40, k=12) and VGG-19 with a momentum optimizer. Specifically, we used a momentum of $0.9$, a batch size of $128$, a dropout of $0.1$, and batch normalization. *Prestopping* has only one unique hyperparameter, the history length $q$, and it was set to be $10$, which was the best value found by the grid search (see Appendix B.1 for details). The hyperparameters used in the compared algorithms were favorably set to be the best values presented in the original papers. As for the training schedule, we trained the network for 120 epochs and used an initial learning rate $0.1$, which was divided by 5 at $50\%$ and $75\%$ of the total number of epochs.

**Algorithms**: We compared *Prestopping* and *Prestopping+* with not only a baseline algorithm but also the *four* state-of-the-art robust training algorithms: *Default* trains the network without any processing for the label noise; *Active Bias* re-weights the backward loss of training samples based on their prediction variance; *Co-teaching* selects a certain number of small-loss samples to train the network based on the co-training (Blum & Mitchell, 1998); *Co-teaching+* is similar to *Co-teaching*, but its small-loss samples are selected from the disagreement set; *SELFIE* selectively refurbishes noisy samples and exploits them together with the small-loss samples. All the algorithms were implemented

---

[4]https://www.cs.toronto.edu/~kriz/cifar.html

[5]https://dm.kaist.ac.kr/datasets/animal-10n

[6]https://kuanghuei.github.io/Food-101N

[7]In FOOD-101N, we used a subset of the entire training data marked with whether the label is correct or not.

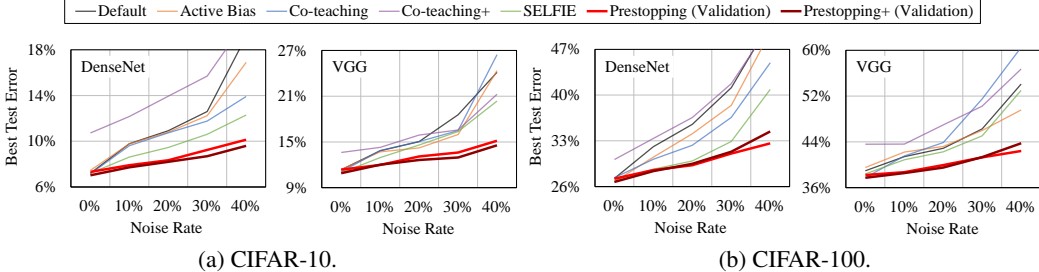

(a) CIFAR-10.  (b) CIFAR-100.

Figure 5: Best test errors using two CNNs on two data sets with varying **pair** noise rates.

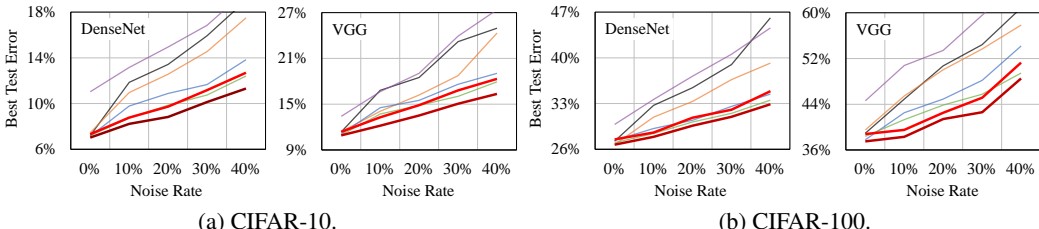

(a) CIFAR-10.  (b) CIFAR-100.

Figure 6: Best test errors using two CNNs on two data sets with varying **symmetric** noise rates.

using TensorFlow 1.8.0 and executed using 16 NVIDIA Titan Volta GPUs. For reproducibility, we provide the source code at `https://bit.ly/2l3g9Jx`. [R2:] In support of reliable evaluation, we repeated every task *thrice* and reported the average and standard error of the best test error, which was evaluated using the network at the time of the lowest validation error. Thus, the clean validation set was used to select the best model in *all* algorithms during the training procedure.

## 5.1 PERFORMANCE COMPARISON

Figures 5 and 6 show the test error of the seven training methods using two CNNs on two data sets with varying *pair* and *symmetric* noise rates, respectively. The results of *Prestopping* and *Prestopping+* in Section 5 were obtained using the *validation* heuristic. See Appendix A for the results of the *noise-rate* heuristic. In order to highlight the improvement of *Prestopping* and *Prestopping+* over the other methods, their lines are dark colored. Figure 7 shows the test error on two *real-world* noisy data sets with different noise rates. The best test errors are detailed in Tables 1 and 2 of Appendix C. [RA:] In addition, more experimental results on Tiny-ImageNet and Clothings (Xiao et al., 2015) data sets are discussed in Appendix B.2 and B.3.

### 5.1.1 RESULT WITH PAIR NOISE (FIGURE 5)

The performance trend in the two network architectures was similar with each other. In general, either *Prestopping* or *Prestopping+* achieved the lowest test error in a wide range of noise rates on both CIFAR data sets. With help of the refurbished samples, *Prestopping+* achieved a slightly better performance than *Prestopping* in CIFAR-10. However, an opposite trend was observed in CIFAR-100; this phenomenon was due to a large number of falsely corrected labels under the pair noise, especially when the number of classes is large (see Appendix B.4 for details). Although *SELFIE* achieved relatively lower test error among the existing methods, the test error of *SELFIE* was still worse than that of *Prestopping*. *Co-teaching* did not work well because many false-labeled samples were misclassified as clean ones; *Co-teaching+* was shown to be even worse than *Co-teaching* despite it being an improvement of *Co-teaching* (see Section 5.4 for details). The test error of *Active Bias* was not comparable to that of *Prestopping*. The performance improvement of ours over the others increased as the label noise became heavier. In particular, at a heavy noise rate of $40\%$, *Prestopping* or *Prestopping+* significantly reduced the *absolute* test error by $2.2pp$–$18.1pp$ compared with the other robust methods.

### 5.1.2 RESULT WITH SYMMETRIC NOISE (FIGURE 6)

Similar to the pair noise, both *Prestopping* and *Prestopping+* generally outperformed the other methods. In particular, the performance of *Prestopping+* was the best at any noise rate on all data sets, because the synergistic effect was higher in symmetric noise than in pair noise. Quantitatively, at a heavy noise rate of $40\%$, our methods showed significant reduction in the *absolute* test error

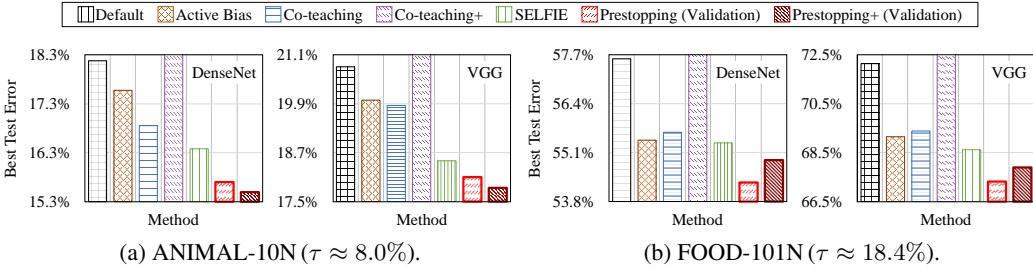

(a) ANIMAL-10N ($\tau \approx 8.0\%$).      (b) FOOD-101N ($\tau \approx 18.4\%$).

Figure 7: Best test errors using two CNNs on two data sets with **real-world** noises.

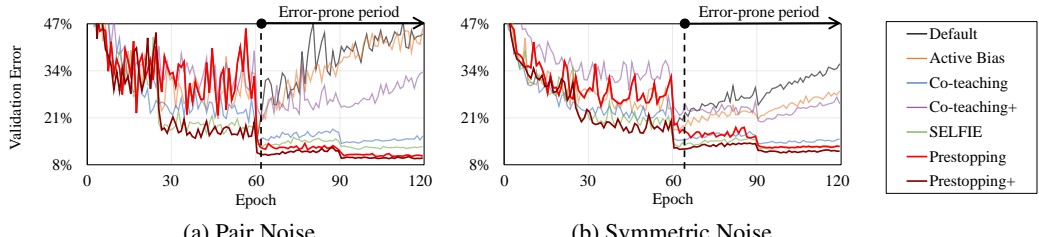

(a) Pair Noise.      (b) Symmetric Noise.

Figure 8: Convergence curves of DenseNet (L=40, k=12) on CIFAR-10 with a noise rate of $40\%$.

by $0.3pp$–$17.5pp$ compared with the other robust methods. Unlike the pair noise, *Co-teaching* and *SELFIE* achieved a low test error comparable to *Prestopping* because true-labeled samples could be well separated from false-labeled ones by their small-loss criteria; hence, the dominance between *Co-teaching* and *Active Bias* was reversed so that the former slightly outperformed the latter.

### 5.1.3 RESULT WITH REAL-WORLD NOISE (FIGURE 7)

Both *Prestopping* and *Prestopping+* maintained their dominance over the other methods under *real-world* label noise as well. *Prestopping+* achieved the lowest test error when the number of classes is small (e.g., ANIMAL-10N), while *Prestopping* was the best when the number of classes is large (e.g., FOOD-101N) owing to the difficulty in label correction of *Prestopping+*. (Thus, practitioners are recommended to choose between *Prestopping* and *Prestopping+* depending on the number of classes in hand.) Specifically, they improved the *absolute* test error by $0.4pp$–$4.6pp$ and $0.5pp$–$8.2pp$ in ANIMAL-10N and FOOD-101N, respectively. Therefore, we believe that the advantage of our methods is unquestionable even in the real-world scenarios.

### 5.2 CONVERGENCE ANALYSIS ON CLEAN VALIDATION DATA (FIGURE 8)

To verify that the error-prone period is eliminated, we plot the convergence of validation error for the seven training methods. The validation error of *Default*, *Active Bias*, and *Co-teaching+* first reached their lowest values before the error-prone period, and then increased rapidly in the error-prone period. On the other hand, the remaining methods kept reducing the validation error even in that period. Notably, *Prestopping+* achieved the lowest validation error at the end, owing to the successful merger of the advanced sample selection in *Prestopping* and the sample refurbishment in *SELFIE*. In addition, the performance of *Prestopping* was shown to be almost the same as and comparable to that of *Prestopping+* in pair and symmetric noises, respectively. The validation error of *SELFIE* also converged faster than those of the other existing methods, but it was still inferior to *Prestopping+*. Thus, these results confirm that our two methods effectively overcome the error-prone period.

### 5.3 ACCURACY OF SAMPLE SELECTION (FIGURE 9)

[RA]: Figure 9 shows the average accuracy of sample selection for two different selection strategies using DenseNet. *Average accuracy* is defined as the average ratio of the true-labeled samples in selected mini-batch samples during the training process. As mentioned earlier in Section 1, because the loss distribution of false-labeled samples was overlapped closely with that of true-labeled ones in pair noise, the average accuracy of the small-loss trick was worse with pair noise than with symmetric noise; it decreased to $75.0\%$ as the noise rate increased in CIFAR-100 with pair noise. On the other hand, the maximal safe set maintained its dominance over the small-loss trick in all data sets regardless of the noise type. In particular, the average accuracy of the maximal safe set was $93.2\%$

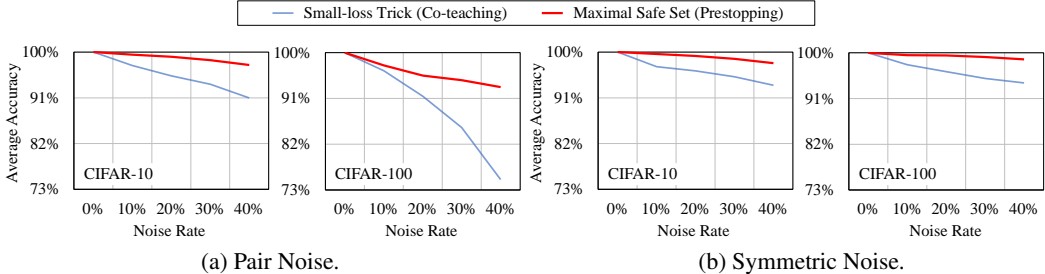

Figure 9: Average accuracy of selecting true-labeled samples by the small-loss trick in *Co-teaching* and the maximal safe set in *Prestopping* using DenseNet on two data sets with synthetic noises.

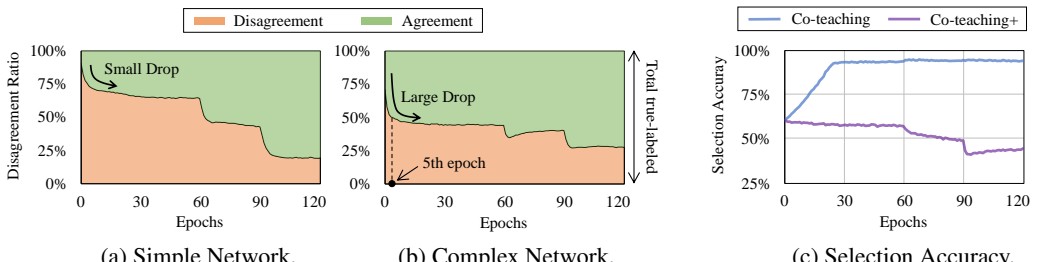

Figure 10: Anatomy of *Co-teaching+* on CIFAR-100 with $40\%$ symmetric noise: (a) and (b) show the change in disagreement ratio for all true-labeled samples, when using *Co-teaching+* to train two networks with different complexity, where "simple network" is a network with seven layers used by Yu et al. (2019), and "complex network" is a DenseNet (L=40, k=12) used for our evaluation; (c) shows the accuracy of selecting true-labeled samples on the DenseNet.

even in CIFAR-100 with pair noise of $40\%$, which led to the huge performance improvement of *Prestopping* over *Co-teaching* in Figure 5(b).

### 5.4 ANATOMY OF *Co-teaching+* (FIGURE 10)

Although *Co-teaching+* is the latest method, its performance was worse than expected, as shown in Section 5.1. Thus, we looked into *Co-teaching+* in more detail. A poor performance of *Co-teaching+* was attributed to the fast consensus of the label predictions for true-labeled samples, especially when training a complex network. In other words, because two complex networks in *Co-teaching+* start making the same predictions for true-labeled samples too early, these samples are excluded too early. As shown in Figures 10(a) and 10(b), the disagreement ratio with regard to the true-labeled samples dropped faster with a complex network than with a simple network, in considering that the ratio drastically decreased to $49.8\%$ during the first 5 epochs. Accordingly, it is evident that *Co-teaching+* with a complex network causes a *narrow exploration* of the true-labeled samples, and the selection accuracy of *Co-teaching+* naturally degraded from $60.4\%$ to $44.8\%$ for that reason, as shown in Figure 10(c). Therefore, we conclude that *Co-teaching+* may not suit a complex network.

## 6 CONCLUSION

In this paper, we proposed a novel two-phase training strategy for the noisy training data, which we call *Prestopping*. The first phase, "early stopping," retrieves an initial set of true-labeled samples as many as possible, and the second phase, "learning from a maximal safe set," completes the rest training process only using the true-labeled samples with high precision. *Prestopping* can be easily applied to many real-world cases because it additionally requires only either a small clean validation set or a noise rate. Furthermore, we combined this novel strategy with sample refurbishment to develop *Prestopping+*. Through extensive experiments using various real-world and simulated noisy data sets, we verified that either *Prestopping* or *Prestopping+* achieved the lowest test error among the seven compared methods, thus significantly improving the robustness to diverse types of label noise. Overall, we believe that our work of dividing the training process into two phases by early stopping is a new direction for robust training and can trigger a lot of subsequent studies.

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

# A  *Prestopping* WITH NOISE-RATE HEURISTIC

Algorithm 2 describes the overall procedure of *Prestopping* with the *noise-rate* heuristic, which is also self-explanatory. Compared with Algorithm 1, only the way of determining the best stop point in Lines 7–9 has changed.

---

**Algorithm 2** *Prestopping* with **Noise-Rate** Heuristic

---

INPUT: $\tilde{\mathcal{D}}$: data, $epochs$: total number of epochs, $q$: history length, $\tau$: noise rate
OUTPUT: $\theta_t$: network parameters
1: $t \leftarrow 1$;  $\theta_t \leftarrow$ Initialize the network parameter;
2: $\theta_{t_{stop}} \leftarrow \emptyset$;  /* The parameter of the stopped network */
3: **for** $i = 1$ **to** $epochs$ **do**  /* **Phase I: Learning from a noisy training data set** */
4:    **for** $j = 1$ **to** $|\tilde{\mathcal{D}}|/|\mathcal{B}_t|$ **do**
5:       Draw a mini-batch $\mathcal{B}_t$ from $\tilde{\mathcal{D}}$;
6:       $\theta_{t+1} = \theta_t - \alpha\nabla\big(\frac{1}{|\mathcal{B}_t|}\sum_{x\in\mathcal{B}_t}\mathcal{L}(x,\tilde{y};\theta_t)\big)$;  /* Update by Eq. (1) */
7:       $train\_err \leftarrow$ Get_Training_Error($\tilde{\mathcal{D}}, \theta_t$);  /* A training error at time $t$ */
8:       **if** $train\_err \leq \tau$ **then**  /*Save the network when $train\_err \leq \tau$*/
9:          $\theta_{t_{stop}} \leftarrow \theta_t$; break;
10:       $t \leftarrow t + 1$;
11: $\theta_t \leftarrow \theta_{t_{stop}}$;  /* Load the network stopped at $t_{stop}$ */
12: **for** $i = stop\_epoch$ **to** $epochs$ **do**  /* **Phase II: Learning from a maximal safe set** */
13:    **for** $j = 1$ **to** $|\tilde{\mathcal{D}}|/|\mathcal{B}_t|$ **do**
14:       Draw a mini-batch $\mathcal{B}_t$ from $\tilde{\mathcal{D}}$;
15:       $\mathcal{S}_t \leftarrow \{x|\text{argmax}_y P(y|x,t;q) = \tilde{y}\}$;  /* A maximal safe set in Definition 3.3 */
16:       $\theta_{t+1} = \theta_t - \alpha\nabla\big(\frac{1}{|\mathcal{S}_t\cap\mathcal{B}_t|}\sum_{x\in\mathcal{S}_t\cap\mathcal{B}_t}\mathcal{L}(x,\tilde{y};\theta_t)\big)$;  /* Update by Eq. (5) */
17:       $t \leftarrow t + 1$;
18: **return** $\theta_t, \mathcal{S}_t$;

---

## A.1  RESULT WITH SYNTHETIC NOISE (FIGURE 11)

To verify the performance of *Prestopping* and *Prestopping+* with the *noise-rate* heuristic in Section 3.2.1, we trained a VGG-19 network on two simulated noisy data sets with the same configuration as in Section 5. Figure 11 shows the test error of our two methods using the noise-rate heuristic as well as those of the other five training methods. Again, the test error of either *Prestopping* or *Prestopping+* was the lowest at most error rates with any noise type. The trend of the noise-rate heuristic here was almost the same as that of the validation heuristic in Section 5.1. Especially when the noise rate was $40\%$, *Prestopping* and *Prestopping+* significantly improved the test error by $5.1pp$–$17.0pp$ in the pair noise (Figure 11(a)) and $0.3pp$–$17.3pp$ in the symmetric noise (Figure 11(b)) compared with the other robust methods.

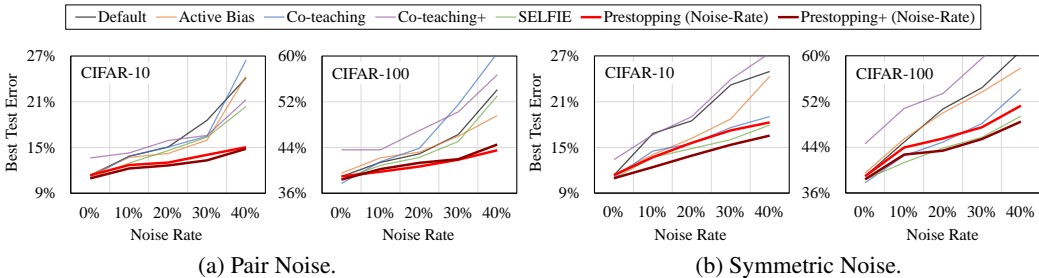

(a) Pair Noise.                    (b) Symmetric Noise.

Figure 11: Best test errors using VGG-19 on two simulated noisy data sets with varying noise rates.

## A.2  COMPARISON WITH VALIDATION HEURISTIC (FIGURE 12)

Figure 12 shows the difference in test error caused by the two heuristics. Overall, the performance with the noise-rate heuristic was worse than that with validation heuristic, even though the worse one outperformed the other training methods as shown in Figure 11. As the assumption of the noise-rate

heuristic does not hold perfectly, a lower performance of the noise-rate heuristic looks reasonable. However, we expect that the performance with this heuristic can be improved by stopping a little earlier than the estimated point, and we leave this extension as the future work.

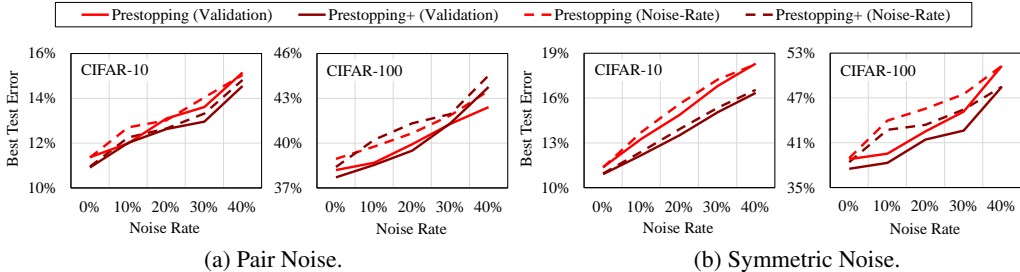

Figure 12: Difference in test error between two heuristics on two simulated noisy data sets.

# B  SUPPLEMENTARY EVALUATION

## B.1  HYPERPARAMETER SELECTION (FIGURE 13)

*Prestopping* requires one additional hyperparameter, the history length $q$ in Definition 3.1. For hyperparameter tuning, we trained DenseNet (L=40, k=12) on CIFAR-10 and CIFAR-100 with a noise rate of $40\%$, and the history length $q$ was chosen in a grid $\in \{1, 5, 10, 15, 20\}$. Figure 13 shows the test error of *Prestopping* obtained by the grid search on two noisy CIFAR data sets. Regardless of the noise type, the lowest test error was typically achieved when the value of $q$ was 10 in both data sets. Therefore, the history length $q$ was set to be 10 in all experiments.

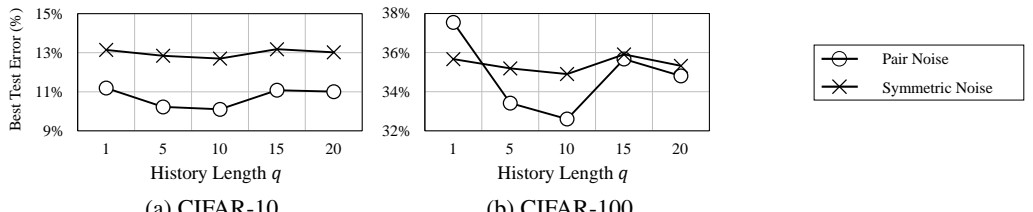

Figure 13: Grid search on CIFAR-10 and CIFAR-100 with two types of noises of $40\%$.

## B.2  EXPERIMENT USING TINY-IMAGENET DATA SET (FIGURE 14)

[R3:] For a larger-scale experiment, we repeated the image classification task on Tiny-ImageNet (200 classes), a subset of ImageNet (Krizhevsky et al., 2012), with $100,000$ training and $10,000$ validation images. Because no test set exists, randomly selected $9,000$ images from the validation set were used as the test set, and the rest $1,000$ validation images were used as the clean validation set for the validation heuristic. The experimental configurations were the same as those in Section 5.

Figure 14 shows the test error of the seven training methods using VGG-19 on Tiny-ImageNet with varying noise rates. Similar to CIFAR data sets in Figures 5 and 6, both *Prestopping* and *Prestopping+* outperformed the other robust methods in both noise types. The only difference was

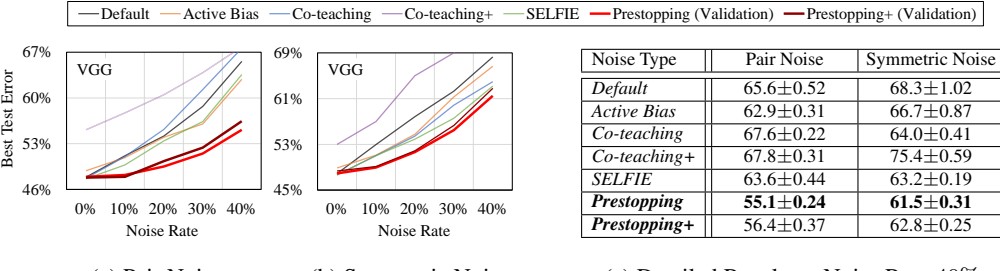

| Noise Type | Pair Noise | Symmetric Noise |
|---|---|---|
| *Default* | 65.6±0.52 | 68.3±1.02 |
| *Active Bias* | 62.9±0.31 | 66.7±0.87 |
| *Co-teaching* | 67.6±0.22 | 64.0±0.41 |
| *Co-teaching+* | 67.8±0.31 | 75.4±0.59 |
| *SELFIE* | 63.6±0.44 | 63.2±0.19 |
| ***Prestopping*** | **55.1±0.24** | **61.5±0.31** |
| ***Prestopping+*** | 56.4±0.37 | 62.8±0.25 |

(a) Pair Noise.   (b) Symmetric Noise.   (c) Detailed Results at Noise Rate $40\%$.

Figure 14: Best test errors using VGG-19 on Tiny-ImageNet with varying noise rates.

that there was no synergistic effect from collaboration with sample refurbishment because of the larger number of classes (i.e., 200). In particular, compared with other robust methods, *Prestopping* showed significant reduction in the *absolute* test error by $7.8pp$–$12.7pp$ at pair noise of $40\%$ and $1.7pp$–$13.9pp$ at symmetric noise of $40\%$.

### B.3    EXPERIMENT USING CLOTHING DATA SET (FIGURE 15)

[R2:] We additionally evaluated the superiority of *Prestopping*(+) on another challenging real-world noisy data set Clothing70k (14 classes), a subset[8] of Clothing1M (Xiao et al., 2015), where its noise rate was estimated at $38.5\%$. For the Clothing70k data set, we randomly selected $70,000$ images from 1 million noisy training images in Clothing1M and used them as its noisy training set; we did exploit the original clean validation and test sets consisting of $14,313$ and $10,526$ images, respectively. The experimental configurations were the same as those in Section 5.

Figure 15 shows the test error of the seven training methods using two CNNs on Clothing70k with real-world noise of $38.5\%$. Overall, *Prestopping* and *Prestopping+* significantly outperformed the other robust methods. In particular, *Prestopping+* achieved the lowest test error because the number of classes was not that large (i.e., 14 classes). Compared with other state-of-the-art methods, it improved the *absolute* test error by $1.1pp$–$3.3pp$ using DenseNet and $0.7pp$–$5.8pp$ using VGG-19.

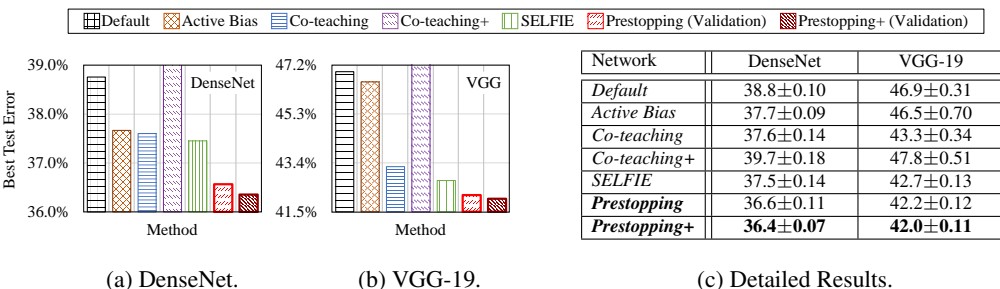

| Network | DenseNet | VGG-19 |
|---|---|---|
| *Default* | 38.8±0.10 | 46.9±0.31 |
| *Active Bias* | 37.7±0.09 | 46.5±0.70 |
| *Co-teaching* | 37.6±0.14 | 43.3±0.34 |
| *Co-teaching+* | 39.7±0.18 | 47.8±0.51 |
| *SELFIE* | 37.5±0.14 | 42.7±0.13 |
| *Prestopping* | 36.6±0.11 | 42.2±0.12 |
| *Prestopping+* | **36.4±0.07** | **42.0±0.11** |

(a) DenseNet.          (b) VGG-19.          (c) Detailed Results.

Figure 15: Best test errors on Clothing70k ($\tau \approx 38.5\%$) along with the detailed result.

### B.4    IMPACT OF NUMBER OF CLASSES ON *Prestopping+* (FIGURE 16)

We investigate the impact of the number of classes on *Prestopping+* under the pair noise of $40\%$. Figure 16 shows the ratio of refurbished samples used for training (hatched bar) with the refurbishing accuracy (solid line) on two data sets with 10 and 100 classes. When the number of classes is small (e.g., 10), *Prestopping+* refurbished most of false-labeled samples very accurately; the refurbishing accuracy was consistently over $91.6\%$ during the entire training period. On the other hand, when the number of classes is large (e.g., 100), the refurbishing accuracy drastically dropped from $92.0\%$ to $76.7\%$ as more samples were refurbished. Thus, as shown in Figure 16(b), collaboration with sample refurbishment is not always synergistic because such a significant drop in the refurbishing accuracy could induce many falsely corrected samples, especially when there are many classes.

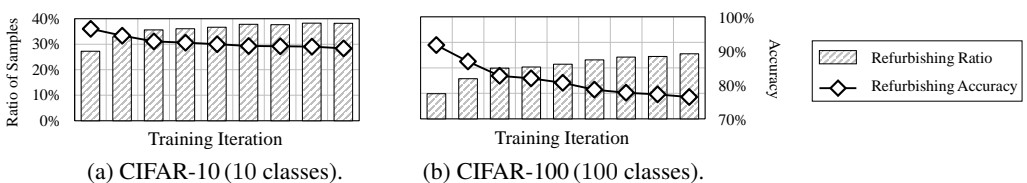

(a) CIFAR-10 (10 classes).          (b) CIFAR-100 (100 classes).

Figure 16: Refurbishing ratio and accuracy of *Prestopping+* when training DenseNet (L=40, k=12) on CIFAR-10 and CIFAR-100 with pair noise of $40\%$.

---

[8][R2:]We used a subset of Clothing1M to finish this experiment within the ICLR'20 rebuttal period.

## C  COMPLETE RESULTS ON BEST TEST ERROR

Table 1 shows the test error of seven training methods using two CNNs on two *simulated* noisy data sets with varying noise rates. Table 2 shows the test error of seven training methods using two CNNs on two *real-world* noisy data sets with different noise rates.

Table 1: The best test errors (%) of seven training methods on two types of **synthetic** noises with varying noise rates (0%, 10%, 20%, 30%, and 40%) in Figures 5 and 6.

| Noise Type | Pair Noise in Figure 5 | | | | Symmetric Noise in Figure 6 | | | |
|---|---|---|---|---|---|---|---|---|
| Data Set | CIFAR-10 | | CIFAR-100 | | CIFAR-10 | | CIFAR-100 | |
| Network | DenseNet | VGG | DenseNet | VGG | DenseNet | VGG | DenseNet | VGG |
| *Default* | 7.20±0.05 | 11.4±0.14 | 27.3±0.35 | 38.9±0.07 | 7.20±0.05 | 11.4±0.14 | 27.3±0.35 | 38.9±0.07 |
| *Active Bias* | 7.45±0.36 | 11.4±0.08 | 26.7±0.48 | 39.5±0.09 | 7.45±0.36 | 11.4±0.08 | 26.7±0.48 | 39.5±0.09 |
| *Co-teaching* | 7.16±0.01 | 11.1±0.14 | 27.4±0.87 | 37.7±0.04 | 7.16±0.01 | 11.1±0.14 | 27.4±0.87 | 37.7±0.04 |
| *Co-teaching+* | 10.7±0.27 | 13.6±0.28 | 30.1±0.24 | 43.6±0.09 | 10.7±0.27 | 13.6±0.28 | 30.1±0.24 | 43.6±0.09 |
| *SELFIE* | 7.20±0.06 | 11.0±0.07 | 27.0±0.41 | 38.4±0.12 | 7.20±0.06 | 11.0±0.07 | 27.0±0.41 | 38.4±0.12 |
| ***Prestopping*** | 7.32±0.06 | 11.4±0.15 | 27.2±0.11 | 38.2±0.17 | 7.32±0.06 | 11.4±0.15 | 27.2±0.11 | 38.2±0.17 |
| ***Prestopping+*** | **7.02±0.11** | **10.9±0.10** | **26.7±0.18** | **37.7±0.16** | **7.02±0.11** | **10.9±0.10** | **26.7±0.18** | 37.5±0.16 |

(a) The best test errors under pair and symmetric noises of 0% ($\tau = 0.0$).

| Noise Type | Pair Noise in Figure 5 | | | | Symmetric Noise in Figure 6 | | | |
|---|---|---|---|---|---|---|---|---|
| Data Set | CIFAR-10 | | CIFAR-100 | | CIFAR-10 | | CIFAR-100 | |
| Network | DenseNet | VGG | DenseNet | VGG | DenseNet | VGG | DenseNet | VGG |
| *Default* | 9.79±0.02 | 13.8±0.62 | 32.1±0.19 | 41.4±0.15 | 11.9±0.74 | 16.8±0.05 | 32.7±0.66 | 44.9±0.24 |
| *Active Bias* | 9.77±0.27 | 13.7±0.14 | 30.5±0.63 | 42.2±0.22 | 11.0±0.44 | 14.1±0.36 | 30.9±0.19 | 45.5±0.21 |
| *Co-teaching* | 9.62±0.09 | 13.9±0.13 | 30.2±0.38 | 41.5±0.20 | 9.79±0.18 | 14.5±0.12 | 29.2±0.15 | 42.5±0.30 |
| *Co-teaching+* | 12.2±0.14 | 14.3±0.07 | 33.4±0.14 | 43.6±0.52 | 13.2±0.34 | 16.7±0.09 | 33.6±0.41 | 50.8±0.24 |
| *SELFIE* | 8.62±0.36 | 13.0±0.07 | 28.7±0.20 | 40.9±0.14 | 8.73±0.12 | 13.7±0.23 | 28.4±0.09 | 41.3±0.32 |
| ***Prestopping*** | 7.91±0.19 | **12.0±0.14** | 28.5±0.14 | 38.7±0.20 | 8.77±0.44 | 13.3±0.03 | 28.6±0.84 | 39.5±0.21 |
| ***Prestopping+*** | **7.72±0.01** | **12.0±0.10** | **28.3±0.12** | **38.5±0.24** | **8.23±0.25** | **12.2±0.13** | **27.9±0.03** | **38.3±0.16** |

(b) The best test errors under pair and symmetric noises of 10% ($\tau = 0.1$).

| Noise Type | Pair Noise in Figure 5 | | | | Symmetric Noise in Figure 6 | | | |
|---|---|---|---|---|---|---|---|---|
| Data Set | CIFAR-10 | | CIFAR-100 | | CIFAR-10 | | CIFAR-100 | |
| Network | DenseNet | VGG | DenseNet | VGG | DenseNet | VGG | DenseNet | VGG |
| *Default* | 10.9±0.44 | 15.1±1.11 | 35.5±0.41 | 42.9±0.21 | 13.4±0.77 | 18.5±0.31 | 35.4±0.79 | 50.7±0.22 |
| *Active Bias* | 10.8±0.17 | 14.2±0.13 | 34.1±0.69 | 43.2±0.09 | 12.6±0.27 | 16.2±0.20 | 33.3±0.25 | 50.0±0.25 |
| *Co-teaching* | 10.7±0.13 | 15.1±0.16 | 32.3±0.95 | 43.9±0.31 | 10.9±0.27 | 15.5±0.08 | 30.4±0.20 | 44.9±0.17 |
| *Co-teaching+* | 13.9±0.38 | 15.9±0.31 | 36.6±0.34 | 47.0±0.34 | 15.0±0.34 | 19.0±0.11 | 37.2±0.18 | 53.4±0.14 |
| *SELFIE* | 9.46±0.10 | 14.6±0.06 | 29.9±0.44 | 42.3±0.17 | 9.86±0.25 | 14.8±0.25 | 30.2±0.37 | 43.9±0.09 |
| ***Prestopping*** | 8.34±0.18 | 13.1±0.08 | **29.3±0.24** | 39.9±0.42 | 9.73±0.28 | 14.9±0.23 | 30.8±0.70 | 42.5±0.20 |
| ***Prestopping+*** | **8.20±0.17** | **12.6±0.03** | 29.5±0.20 | **39.5±0.25** | **8.83±0.30** | **13.5±0.05** | **29.6±0.40** | **41.6±0.08** |

(c) The best test errors under pair and symmetric noises of 20% ($\tau = 0.2$).

| Noise Type | Pair Noise in Figure 5 | | | | Symmetric Noise in Figure 6 | | | |
|---|---|---|---|---|---|---|---|---|
| Data Set | CIFAR-10 | | CIFAR-100 | | CIFAR-10 | | CIFAR-100 | |
| Network | DenseNet | VGG | DenseNet | VGG | DenseNet | VGG | DenseNet | VGG |
| *Default* | 12.6±1.98 | 18.6±1.11 | 41.1±0.43 | 46.3±0.25 | 15.9±1.25 | 23.2±0.41 | 39.0±0.78 | 54.4±0.25 |
| *Active Bias* | 12.2±0.80 | 15.6±0.12 | 38.4±0.91 | 46.0±0.16 | 14.6±0.48 | 18.7±0.31 | 36.7±0.66 | 53.7±0.12 |
| *Co-teaching* | 11.8±0.54 | 16.5±0.11 | 36.6±1.08 | 51.5±0.36 | 11.7±0.35 | 17.6±0.02 | 32.6±0.33 | 48.2±0.34 |
| *Co-teaching+* | 15.7±0.19 | 16.6±0.12 | 41.8±0.79 | 50.2±0.21 | 16.8±0.12 | 24.0±0.17 | 40.6±0.24 | 59.6±0.23 |
| *SELFIE* | 10.6±0.34 | 16.4±0.17 | 32.9±0.13 | 45.1±0.16 | 10.8±0.09 | 16.0±0.24 | 31.5±0.33 | 45.8±0.60 |
| ***Prestopping*** | 9.24±0.24 | 13.6±0.14 | **31.1±0.73** | **41.3±0.14** | 11.2±0.97 | 16.8±0.23 | 32.1±0.94 | 45.2±0.31 |
| ***Prestopping+*** | **8.67±0.29** | **13.0±0.09** | 31.3±0.19 | **41.3±0.05** | **10.1±0.70** | **15.1±0.05** | **31.0±0.88** | **42.6±0.06** |

(d) The best test errors under pair and symmetric noises of 30% ($\tau = 0.3$).

| Noise Type | Pair Noise in Figure 5 | | | | Symmetric Noise in Figure 6 | | | |
|---|---|---|---|---|---|---|---|---|
| Data Set | CIFAR-10 | | CIFAR-100 | | CIFAR-10 | | CIFAR-100 | |
| Network | DenseNet | VGG | DenseNet | VGG | DenseNet | VGG | DenseNet | VGG |
| *Default* | 19.2±0.63 | 24.1±0.97 | 51.2±0.40 | 54.1±0.82 | 19.0±1.66 | 25.0±0.19 | 46.1±0.72 | 60.9±0.08 |
| *Active Bias* | 16.9±0.93 | 24.4±1.24 | 49.6±0.36 | 49.6±0.60 | 17.5±0.90 | 24.3±0.36 | 39.2±0.84 | 57.9±0.20 |
| *Co-teaching* | 13.9±1.11 | 26.5±2.47 | 45.0±1.06 | 60.5±0.69 | 13.8±0.41 | 19.1±0.07 | 34.5±0.13 | 54.2±0.45 |
| *Co-teaching+* | 20.6±0.74 | 21.3±0.20 | 50.7±0.65 | 56.7±0.16 | 19.9±0.28 | 27.4±0.24 | 44.6±1.04 | 66.0±0.31 |
| *SELFIE* | 12.3±0.80 | 20.4±0.24 | 40.9±1.00 | 53.0±0.45 | 12.4±0.07 | 18.0±0.14 | 33.5±0.80 | 49.5±0.07 |
| ***Prestopping*** | 10.1±0.20 | 15.2±0.54 | **32.6±0.40** | **42.4±0.28** | 12.7±0.33 | 18.3±0.07 | 34.9±0.63 | 51.3±0.44 |
| ***Prestopping+*** | **9.60±0.13** | **14.6±0.17** | 34.4±0.10 | 43.8±0.14 | **11.3±0.03** | **16.4±0.08** | **32.9±0.20** | **48.5±0.05** |

(e) The best test errors under pair and symmetric noises of 40% ($\tau = 0.4$).

Table 2: The best test errors (%) on **real-world** noises in Figure 7.

| Data Set | ANIMAL-10N | | FOOD-101N | |
|---|---|---|---|---|
| Network | DenseNet | VGG | DenseNet | VGG |
| *Default* | 18.2±0.15 | 20.8±0.36 | 57.6±0.20 | 72.1±0.45 |
| *Active Bias* | 17.6±0.13 | 20.0±0.30 | 55.4±0.10 | 69.2±0.84 |
| *Co-teaching* | 16.9±0.14 | 19.9±0.01 | 55.6±0.09 | 69.4±0.27 |
| *Co-teaching+* | 19.8±0.11 | 22.4±0.61 | 58.6±0.24 | 75.5±0.53 |
| *SELFIE* | 16.4±0.21 | 18.5±0.14 | 55.4±0.17 | 68.6±0.24 |
| ***Prestopping*** | 15.7±0.18 | 18.1±0.14 | **54.3±0.24** | **67.3±0.21** |
| ***Prestopping+*** | **15.5±0.12** | **17.8±0.07** | 54.9±0.17 | 67.9±0.09 |

# D    CASE STUDY: NOISY LABELS IN ORIGINAL CIFAR-100

One interesting observation is a noticeable improvement of *Prestopping+* even when the noise rate was $0\%$, as shown in Table 1(a). It was turned out that *Prestopping+* sometimes refurbished the labels of the false-labeled samples which were *originally* contained in the CIFAR data sets. Figure 17 shows a few successful refurbishment cases. For example, an image falsely annotated as a "Boy" was refurbished as a "Baby" (Figure 17(a)), and an image falsely annotated as a "Mouse" was refurbished as a "Hamster" (Figure 17(c)). Thus, this sophisticated label correction of *Prestopping+* helps overcome the residual label noise in well-known benchmark data sets, which are misconceived to be clean, and ultimately further improves the generalization performance of a network.

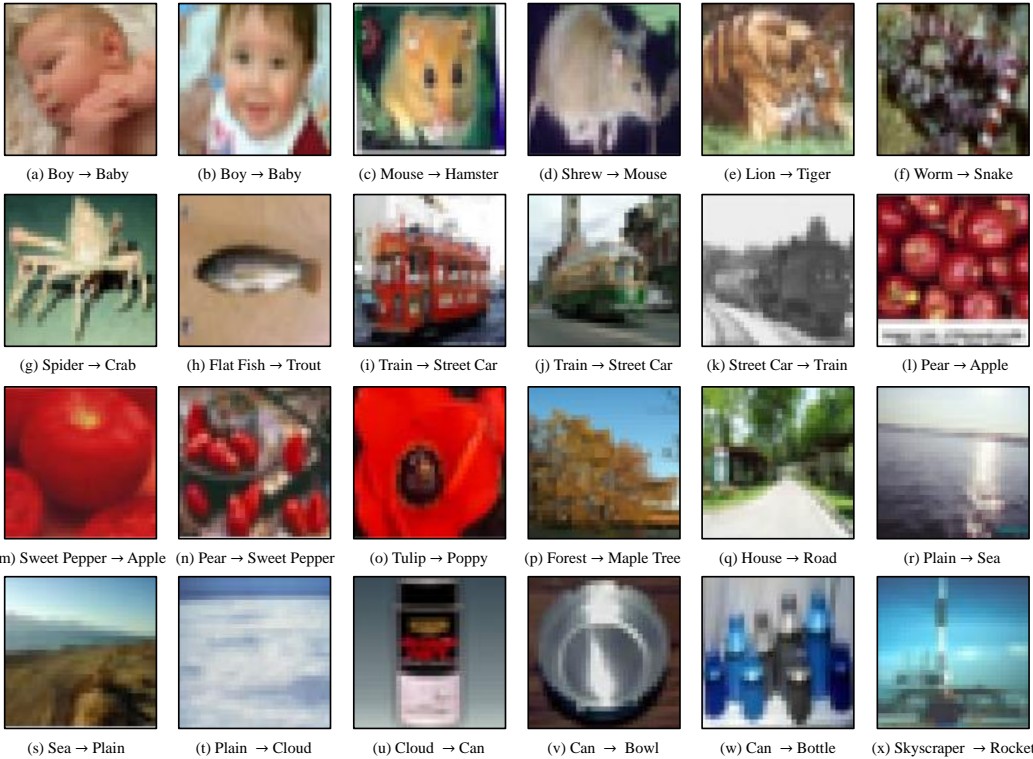

(a) Boy → Baby    (b) Boy → Baby    (c) Mouse → Hamster    (d) Shrew → Mouse    (e) Lion → Tiger    (f) Worm → Snake

(g) Spider → Crab    (h) Flat Fish → Trout    (i) Train → Street Car    (j) Train → Street Car    (k) Street Car → Train    (l) Pear → Apple

(m) Sweet Pepper → Apple    (n) Pear → Sweet Pepper    (o) Tulip → Poppy    (p) Forest → Maple Tree    (q) House → Road    (r) Plain → Sea

(s) Sea → Plain    (t) Plain → Cloud    (u) Cloud → Can    (v) Can → Bowl    (w) Can → Bottle    (x) Skyscraper → Rocket

Figure 17: Refurbishing of false-labeled samples *originally* contained in CIFAR-100. The subcaption represents "original label" → "refurbished label" recognized by *Prestopping+*.

