# OpenReview forum: "Prestopping: How Does Early Stopping Help Generalization Against Label Noise?"
_ICLR.cc/2020/Conference — Reject_

### Official Review · AnonReviewer2 · 2019-10-20
**Official Blind Review #2**

**Rating:** 3

**Review:**

The paper proposes to study how early stopping in optimization helps find confident examples. Overall, the paper is well-organized and easy to read. Although there is some parallel study regarding the theoretical aspect of how early stopping help finds confident examples (i.e., Gradient Descent with Early Stopping is Provably Robust to Label Noise for Overparameterized Neural Networks, which has unfortunately not been cited), the paper focuses on the empirical perspective. A thorough empirical study illustrating how early stop works would interest the label noise community.

The authors claim that early stopping is efficient to find a maximal safe set. I think it would be necessary to illustrate the maximal safe set for all the datasets. The authors only did this for one case of CIFAR-100, which is not convincing. The small loss based learning has shown the confident examples extracted. It would be essential to compare the proposed method with those methods.

The authors are suggested to compare the proposed method with more baselines. There are lots of algorithms exploiting the transition matrix and with statistically consistent estimators. The authors have ignored all those methods.

It is unclear from the paper that if the baselines have used the clean validation sets. For fair comparison, those clean data should be used in the training procedures of the baselines.

Clothing1M is a more challenging dataset with real-world label noise. The dataset also includes some clean data for validation use.  The authors should verify the effectiveness of the proposed method on this dataset.

The authors are suggested to make it clear why noise rates are sometimes available for use.

**Experience Assessment:**

I have published in this field for several years.

**Review Assessment: Checking Correctness Of Derivations And Theory:**

I carefully checked the derivations and theory.

**Review Assessment: Checking Correctness Of Experiments:**

I assessed the sensibility of the experiments.

**Review Assessment: Thoroughness In Paper Reading:**

I read the paper thoroughly.

---

> ### Author Response · Authors · 2019-11-11
> **Response to Reviewer 2 (1/2)**
>
> Thank you for raising your insightful and detailed comments. We have revised our paper to address your concerns; please see the modified parts marked as “R2” in teal color and marked as “RA” in magenta color.
>
> Below is the summary of our response to your concerns:
>
> Q2-1. "Gradient Descent with Early Stopping is Provably Robust to Label Noise for Overparameterized Neural Networks" is not cited.
> A2-1. The suggested paper provides a theoretical analysis of early stopping and makes the claim that the early stopped network is fairly robust to the label noise compared with the fully trained one. However, please note that this paper does not provide any idea to exploit the early stopped network for "sample selection". In contrast, Prestopping adopts early stopping to derive a seed for confident samples called the maximal safe set. Thus, our novelty lies in the “merger” of early stopping and learning from the maximal safe set. This merger is our unique contribution. We have highlighted this contribution in Section 2.
>
> Q2-2. It would be necessary to illustrate the maximal safe set for all the data sets. Also, it would be essential to compare the proposed method with small loss-based methods.
> A2-2. Thank you for your careful comments. We have added the results in Figure 4 (Section 3.2.2). The label precision and recall were very high in all datasets. Furthermore, we have added the performance evaluation of selecting clean samples using the small-loss trick (Co-teaching) and the maximal safe set (Prestopping). The selection accuracy of Prestopping was much higher (by up to 18.2pp) than that of Co-teaching (See Section 5.3 for details).
>
> Q2-3. The authors are suggested to compare the proposed method with more baselines.
> A2-3. Thank you for your comments. We could not add more baselines during the rebuttal period. However, many baselines exploiting the transition matrix (e.g., S-model[1], F-correction[2]) have been reported to perform poorly than Co-teaching [3] we included, and we confirm that Prestopping outperforms more recent baselines according to the accuracy reported in the literature. For example, for CIFAR-10 and 30%-pair noise, the test accuracy of Prestopping was 90.8%, while the test accuracy of ITLM[4] was 88.2%; For CIFAR-10 and 40%-symmetric noise, the test accuracy of Prestopping was 88.7%, while the test accuracy of D2L[5] was 83.4%. We will definitely add more baselines to the camera-ready version, if accepted.
>
> [1] Training Deep Neural Networks using A Noise Adaptation Layer (ICLR’17)
> [2] Making Deep Neural Networks Robust to Label Noise: A Loss Correction Approach (CVPR’17)
> [3] Co-teaching: Robust Training of Deep Neural Networks with Extremely Noisy Labels (NIPS’18)
> [4] Learning with Bad Training Data via Iterative Trimmed Loss Minimization (ICML’19)
> [5] Dimensionality-Driven Learning with Noisy Labels (ICML’18)
>
> Q2-4. It is unclear from the paper that if the baselines have used the clean validation sets.
> A2-4. For fair comparison, all the baselines used the clean validation sets to select the best number of epochs based on the validation accuracy. The best test error reported in our paper was evaluated using the network at the time of the lowest validation error. We have clarified this issue in the updated paper. (See Section 5 for details).
>
> Q2-5. Clothing1M is a more challenging dataset with real-world label noise.
> A2-5. We have conducted additional experiments for Clothing1M. Prestopping significantly outperformed the existing methods by 0.7—5.8pp. Please see Appendix B.3 of the updated paper.
>
> Q2-6. The authors are suggested to make it clear why noise rates are sometimes available for use.
> A2-6. The noise rate is typically not known in real-world scenarios. Thus, the validation heuristic is preferred to the noise-rate heuristic. However, users may figure out the noise rate by manually investigating a small, random sample of a data set. It is also available to use the estimated noise rate or noise transition matrix based on the recent work[1, 2, 6].
> [6] Using Trusted Data to Train Deep Networks on Label Corrupted by Severe Noise (NIPS'18)
>
> If you want to know the other changes in the updated paper, see the comment "Summary of the overall revision" at the top.

---

> ### Author Response · Authors · 2019-11-14
> **Response to Reviewer 2 (2/2)**
>
> We were very happy to have the opportunity to reflect your insightful comments. During the remaining rebuttal period, we are willing to reflect your additional comments if you have. Thank you again for your valuable comments.

---

### Official Review · AnonReviewer1 · 2019-10-23
**Official Blind Review #1**

**Rating:** 3

**Review:**

This paper proposes a two-phase training method for learning with label noise.

On the positive side, this paper focuses on the idea of prestopping and proposes several relevant definitions to formalize their idea and come up with a heuristic algorithm.

However, I believe the paper has missed several very relevant papers that provides very similar ideas. Both [2] & [3] provide theoretical analysis to why early stopping matters in learning with label noise for DNNs. Before these two papers, [1] also observed that the learning trajectories for clean and noisy samples are different in label noise problem, and they used early stopping in their experiments to address this issue. Given these existing literatures, the contribution of this paper should be considered more properly.

[1] Learning with Bad Training Data via Iterative Trimmed Loss Minimization, Yanyao Shen, Sujay Sanghavi, ICML 2019.
[2] Hu, Wei, Zhiyuan Li, and Dingli Yu. "Understanding Generalization of Deep Neural Networks Trained with Noisy Labels." arXiv preprint arXiv:1905.11368 (2019).
[3] Li, Mingchen, Mahdi Soltanolkotabi, and Samet Oymak. "Gradient descent with early stopping is provably robust to label noise for overparameterized neural networks." arXiv preprint arXiv:1903.11680 (2019).


**Experience Assessment:**

I have published one or two papers in this area.

**Review Assessment: Checking Correctness Of Derivations And Theory:**

I assessed the sensibility of the derivations and theory.

**Review Assessment: Checking Correctness Of Experiments:**

I assessed the sensibility of the experiments.

**Review Assessment: Thoroughness In Paper Reading:**

I read the paper at least twice and used my best judgement in assessing the paper.

---

> ### Author Response · Authors · 2019-11-11
> **Response to Reviewer 1 (1/2)**
>
> Thank you for raising your insightful and detailed comments. We have revised our paper to address your concerns; please see the modified parts marked as “R1” in brown color and marked as “RA” in magenta color.
>
> Below is the summary of our response to your concerns:
>
> Q1-1: The reviewer believes the paper has missed several very relevant papers that provide very similar ideas.
> A1-1: We fully understand your concern. However, please note that, in our work, early stopping is adopted only to derive a "seed” for the maximal safe set. Our novelty lies in the "merger” of early stopping and learning from the maximal safe set. On the other hand, the two suggest papers [2, 3] provide only empirical or theoretical evidence that the early stopped network is fairly robust to label noise compared with the fully trained one.
>
> As for the rest work [1], although they used early stopping in their experiment, they did not answer the following challenging question: when is the best point for the early stop? They simply stopped the network at a certain point and then used small-loss samples as clean samples. In our work, we thoroughly explore the best stop point to obtain not only quantitatively sufficient but also qualitatively less noisy "seed” for the clean samples (i.e., maximal safe set). The advantage of our criterion for the safe set is proven by the fact we exploit almost all true-labeled samples in "both” types of noise (See Figure 4). Furthermore, we have added the performance evaluation of selecting clean samples using the small-loss trick and the maximal safe set. The selection accuracy using the maximal safe set was much higher (by up to 18.2pp) than that using the small-loss trick. Please see Section 5.3 for details (This section has been added at the request of Reviewer 2).
>
> Robust training of DNNs with label noise is a very active research topic. Thus, many studies are being conducted "in parallel" along this direction. Thus, it seems possible that well-established techniques (e.g., early stopping) could be shared in these parallel studies. Nevertheless, the above-mentioned merger is our unique contribution. We have clarified this contribution to our updated paper (See Section 2 for details).
>
> Besides, Prestopping has achieved the "best accuracy" in popular benchmark data sets, as far as we know. We strongly believe this high accuracy is the most important contribution. Specifically, Prestopping is shown to outperform Co-teaching+(ICML’19), an improvement of Co-teaching. Although we could not directly compare Prestopping with ITLM[1], for CIFAR-10 and 30%-pair noise, the test accuracy of Prestopping was 90.8%, while the test accuracy of ITLM[1] was 88.2%.
>
> If you want to know the other changes in the updated paper, see the comment "Summary of the overall revision" at the top.

---

> ### Author Response · Authors · 2019-11-14
> **Response to Reviewer 1 (2/2)**
>
> We were very happy to have the opportunity to reflect your insightful comments. During the remaining rebuttal period, we are willing to reflect your additional comments if you have. Thank you again for your valuable comments.

---

### Official Review · AnonReviewer4 · 2019-10-30
**Official Blind Review #4**

**Rating:** 6

**Review:**

This paper presents a training approach on label noise datasets and outperforms state-of-art methods. It defines the samples whose average probability on assigned label in recent q iterations is largest among all labels as memorized samples, in the sense of the network memorize these samples. Then authors proposed two stage method which firstly early-stops at minimum validation error (or $\tau$ memorized rate), and then trains on maximal safe set that gathers memorized samples. The experiments compared several state-of-art approaches and showed that the proposed method benefits from early-stopping and safe set. Authors also showed that the prestopping idea can also be used to improve other approaches.

Pros:

The proposed method achieves better performance than state-of-art methods.

Authors have good experiments which evaluate on multiple datasets and algorithms.

Authors also investigate the relation between model complexity and performance of co-teaching+

Cons:

Many recent papers indicate the “error-prone period”, authors should include related works about early-stopping on label noise training.
https://arxiv.org/pdf/1901.09960.pdf fig1
https://arxiv.org/pdf/1903.11680.pdf fig5
https://arxiv.org/pdf/1906.05392.pdf fig3

Although the method achieves good performance, since the idea is a bit straightforward especially after exploring above papers, I am slightly worried about novelty of the ideas.



**Experience Assessment:**

I have read many papers in this area.

**Review Assessment: Checking Correctness Of Derivations And Theory:**

I carefully checked the derivations and theory.

**Review Assessment: Checking Correctness Of Experiments:**

I carefully checked the experiments.

**Review Assessment: Thoroughness In Paper Reading:**

I read the paper thoroughly.

---

> ### Author Response · Authors · 2019-11-11
> **Response to Reviewer 4 (1/2)**
>
> Thank you for raising your insightful and detailed comments. We have revised our paper to address your concerns; please see the modified parts marked as “RA” in magenta color.
>
> Below is the summary of our response to your concerns:
>
> Q4-1. The authors should include related work about early-stopping on label noise training.
> A4-1. Thank you for suggesting useful references. The suggested papers provide empirical or theoretical evidence that the early stopped network is fairly robust to label noise compared with the fully trained one. However, these papers do not provide any idea to exploit the early stopped network for "sample selection". In contrast, Prestopping adopts early stopping to derive a seed for clean samples called the maximal safe set. Also, Prestopping is clearly different from recent studies that select the clean samples based on the small-loss[1-5], in considering that memorized samples are regarded as the maximal safe set. Thus, our novelty lies in the “merger” of early stopping and learning from the maximal safe set. This is our unique contribution. We have highlighted this contribution by including the suggested papers (See Section 2 for details).
>
> [1] Co-teaching: Robust Training of Deep Neural Networks with Extremely Noisy Labels (NIPS’18)
> [2] MentorNet: Learning Data-driven Curriculum for Very Deep Neural Networks on Corrupted Labels (ICML'18)
> [3] Understanding Generalization of Deep Neural Networks Trained with Noisy Labels (ICML'19)
> [4] Learning with Bad Training Data via Iterative Trimmed Loss Minimization (ICML'19)
> [5] How does Disagreement Help Generalization against Label Corruption? (ICML'19)
>
> Q4-2. The reviewer is slightly worried about novelty of the ideas.
> A4-2. We understand your concern. However, as mentioned earlier, our novelty lies in selecting highly confident samples initially derived from the early stopped network. Owing to the advantage of the merger, Prestopping has achieved much higher test accuracy than state-of-the-art methods, and Prestopping is the best performer, as far as we know. Furthermore, the accuracy of selecting true-labeled samples via Prestopping was much higher (by up to 18.2pp) than that via the small-loss trick. Please see Section 5.3 for details (This section has been added at the request of Reviewer 2). We hope that this significant contribution is acknowledged.
>
> If you want to know the other changes in the updated paper, see the comment "Summary of the overall revision" at the top.

---

> ### Author Response · Authors · 2019-11-14
> **Response to Reviewer 4 (2/2)**
>
> We were very happy to have the opportunity to reflect your insightful comments. During the remaining rebuttal period, we are willing to reflect your additional comments if you have. Thank you again for your valuable comments.

---

### Official Review · AnonReviewer3 · 2019-10-30
**Official Blind Review #3**

**Rating:** 3

**Review:**

This paper proposes a training strategy for robustness against label noise. The training strategy is simple and straightforward. The neural network will first be trained on the entire dataset with all the noisy labels. After obtaining the network with lowest validation error, the network will be used to make a prediciton on the original training set and select a subset of it to construct a maximal safe set. Finally, the network will be findtuned on this maximal safe set. The training strategy is very similar to tradictional  self-training in semi-superivsed learning and co-training for domain adaptation ([Co-training for domain adaptation, NIPS 2011]), except that the proposed prestopping only iterate the procedure once.

The paper discusses two important questions for the method: (1) when to early stop the training; (2) how to constuct a maximal safe set. The authors' responses to these questions are very natual but less interesting. Using the lowest validation error to early stop the training could be suboptimal, since the small validation set can not fully capture the data distribution and could make the network empirically overfit to this validation set. The criterion to contruct a maxial safe set is also conventional, and is similar to what a number of papers are doing, for example,
[1] Co-training for domain adaptation, NIPS 2011
[2] Self-ensembling for visual domain adaptation, ICLR 2018
[3] A dirt-t approach to unsupervised domain adaptation, ICLR 2018
[4] Iterative learning with open-set noisy labels, CVPR 2018

In experiments, the results are not very surprising. There are some baselines that adopt a similar (iterative) pipeline (learning the network - selecting a subset of the training samples - re-learning the network):
[1] Iterative Learning with Open-set Noisy Labels, CVPR 2018
[2] Dimensionality-Driven Learning with Noisy Labels, ICML 2018
[3] Symmetric Cross Entropy for Robust Learning with Noisy Labels, ICCV 2019
The authors can consider to compare to some of these baselines, especially [1] and [2]. The difference between the paper and [1,2] is basically the criterion to construct the maximal safe subset.

Besides, I suggest the authors to conduct large-scale experiments on ImageNet or even a subset of ImageNet, since the difficulty of detecting label noise is much higher when the resolution of images become bigger. CIFAR-10 and CIFAR-100 only contain 32x32 images, which is far less challenging.

Overall, I think the paper is well written, the idea is clearly presented, and the experiments also seem convinceing. However, the contribution of this paper is very incremental.

**Experience Assessment:**

I have read many papers in this area.

**Review Assessment: Checking Correctness Of Derivations And Theory:**

I assessed the sensibility of the derivations and theory.

**Review Assessment: Checking Correctness Of Experiments:**

I assessed the sensibility of the experiments.

**Review Assessment: Thoroughness In Paper Reading:**

I read the paper at least twice and used my best judgement in assessing the paper.

---

> ### Author Response · Authors · 2019-11-11
> **Response to Reviewer 3 (1/2)**
>
> Thank you for raising your insightful and detailed comments. We have revised our paper to address your concerns; please see the modified parts marked as “R3” in violet color and marked as “RA” in magenta color.
>
> Below is the summary of our response to your concerns:
>
> Q3-1. The criterion to construct a maximal safe set is conventional.
> A3-1. We understand your point. The overall framework is similar to self-training or co-training, as you mentioned. Nevertheless, the detail of deciding safe samples is obviously different. The advantage of our criterion is proven by the fact we exploit almost all true-labeled samples in "both” types of noise (See Figure 4). Then, this advantage leads to much higher test accuracy (See Q3-3 & Q3-4).
>
> Q3-2. The pipeline (learning the network - selecting a subset of the training samples - re-learning the network) has been used in other methods.
> A3-2. Yes, you are right. The existing methods which belong to the "sample selection" category in Section 2 have the same pipeline, including the most recent approach Co-teaching+ (ICML’19). In this sense, we believe that the novelty can be found in how to select clean samples.
>
> Specifically, for sample selection, Iterative[1] used the local outlier factor algorithm and D2L[2] used the local intrinsic dimensionality. Also, there are many studies used the small-loss trick such as Co-teaching (ICML'18), Co-teaching+ (ICML'19), ITLM (ICML'19), and INCV (ICML'19). Compared with these methods, our novelty lies in selecting highly safe samples called the maximal safe set initially derived from the early stopped network. We have clarified the difference between sample selection methods by including the recent work we missed (See Section 2 for details).
>
> Q3-3. More baselines need to be compared.
> A3-3. Thank you for your comments. We could not add the two methods[1,2] during the rebuttal period, but we could indirectly compare Prestopping with Iterative[1] and D2L[2] because the two papers have the common setting. Specifically, Prestopping outperforms Co-teaching and Co-teaching+, but Iterative[1] has been reported to perform poorly than them. In addition, according to the test accuracy in the literature, the test accuracy of Prestopping was 88.7% for CIFAR-10 with 40%-symmetric noise, while the test accuracy of D2L[2] was 83.4%. We will definitely add more baselines to the camera-ready version, if accepted.
>
> Q3-4. ImageNet or even a subset of ImageNet needs to be used for experiments.
> A3-4. We have included Tiny-ImageNet in the updated experiments. Prestopping maintained dominance also in Tiny-ImageNet by 1.7—13.9pp. Please see Appendix B.2 of the updated paper.
> Besides, ANIMAL-10N and Food-101N datasets contain 64x64 images, and a new challenging real-world dataset called Clothing1M have been added in Appendix B.3.
>
> Q3-5. The contribution of this paper is very incremental.
> A3-5. We understand your concern. However, as mentioned earlier, owing to the advantage of our maximal safe set, Prestopping has achieved much higher test accuracy than the state-of-the-art methods, and Prestopping is the best performer, as far as we know. Furthermore, the accuracy of selecting true-labeled samples via Prestopping was much higher (by up to 18.2pp) than that via the small-loss trick. Please see Section 5.3 for details (This section has been added at the request of Reviewer 2). We hope that this significant contribution is acknowledged. Thank you for your insightful comments.
>
> If you want to know the other changes in the updated paper, see the comment "Summary of the overall revision" at the top.

---

> ### Author Response · Authors · 2019-11-14
> **Response to Reviewer 3 (2/2)**
>
> We were very happy to have the opportunity to reflect your insightful comments. During the remaining rebuttal period, we are willing to reflect your additional comments if you have. Thank you again for your valuable comments.

---

### Author Response · Authors · 2019-09-27
**Typo in Eq. (6).**

We apology to the readers for the typo in Eq. (6).
- $\{x \in \mathcal{R}_{t} \cap \mathcal{S}_{t_{end}}\}$ -> $\{x \in \mathcal{R}_{t} \cup \mathcal{S}_{t_{end}}\}$

---

### Public Comment · ~Dan_Hendrycks1 · 2019-10-12
**Related Work**

Hi,

The theoretical work _Gradient Descent with Early Stopping is Provably Robust to Label Noise for Overparameterized Neural Networks_ seems quite related and should probably be mentioned. https://arxiv.org/pdf/1903.11680.pdf

The following empirical work uses quick convergence in order to combat label noise memorization.
_Using Pre-Training Can Improve Model Robustness and Uncertainty_ https://arxiv.org/pdf/1901.09960.pdf (ICML 2019)
Instead of "pre-stopping", pre-training allows quick convergence which lets one side-step the noise memorization phase. This work is also worth mentioning. (See "By training for longer, the network eventually begins to model and memorize label noise" and Section 3.2)

---

> ### Author Response · Authors · 2019-10-13
> **Thanks for your comments.**
>
> Thanks a lot for your interest and comments on the two recent work we missed. We will cover both papers during the rebuttal period.
>
> We think that the first work you mentioned will give many theoretical intuitions to improve our two heuristics for the question "(Q1) When is the best point to early stop the training process?". Actually, we were looking for a theoretical paper related to this question for our future research. Besides, please note that the early stopping is adopted to derive a "maximal safe set" that enables noise-free training during the remaining learning period. The significant performance improvement of our method is achieved by a merger of "early stopping" and "learning form the maximal safe set".
>
> Also, the second work, one another direction to use "quick convergence by pre-training", is really interesting!
> Following the paper, the potential of the weakly pre-trained model seems to be great for a wide range of tasks such as adversarial perturbations and label corruption. Thus, it is likely that our method "Prestopping" can be further enhanced by taking advantage of a weakly pre-trained model for publicly available data (e.g., ImageNet).

---

### Author Response · Authors · 2019-11-11
**Summary of the overall revision**

Dear reviewers, we would like to thank for your insightful and detailed comments.
We have responded individually to each reviewer and below is the summary of the overall revision of the updated paper.

1.	*** Contribution of the paper (Reviewer 1, 2, 3, 4) ***
Most reviewers agreed that the paper is well written, the idea is clearly presented, and the experiments also seem convincing. However, because we missed many relevant papers, all reviewers pointed out that the novelty of the paper is unclear.

As pointed out by the reviewers, few recent studies[1-3] provide empirical or theoretical evidence that the early stopped network is fairly robust to label noise compared with the fully trained one. However, they do not provide any idea to exploit early stopping for "sample selection", which is one of the common directions for handling label noise. In contrast, our work adopts early stopping to derive a "seed” for the safe samples called the maximal safe set and explores the best stop point to obtain not only quantitatively sufficient but also qualitatively less noisy "seed” for the maximal safe set.

[1] Gradient Descent with Early Stopping is Provably Robust to Label Noise for Overparameterized Neural Networks (arXiv'19)
[2] Understanding Generalization of Deep Neural Networks Trained with Noisy Labels (ICML'19)
[3] Using Pre-training can Improve Model Robustness and Uncertainty (ICML’19)

Compared with the work [2, 4-8] belonging to "sample selection” category, our novelty lies in how to select clean samples. Specifically, for sample selection, Iterative[4] used the local outlier factor algorithm and D2L[5] used the local intrinsic dimensionality. Also, INCV[2], Co-teaching[6], Co-teaching+[7], and ITLM[8] used the small-loss trick. Please note, in our paper, Prestopping uses the maximal safe set derived from the memorized samples at the (estimated) best stop point. Definitely, the criterion of the maximal safe set is different.

[4] Iterative Learning with Open-set Noisy Labels, (CVPR’18)
[5] Dimensionality-Driven Learning with Noisy Labels (ICML’18)
[6] Co-teaching: Robust Training of Deep Neural Networks with Extremely Noisy Labels (NIPS’18)
[7] How does Disagreement Help Generalization against Label Corruption? (ICML'19)
[8] Learning with Bad Training Data via Iterative Trimmed Loss Minimization (ICML’19)

Robust training of DNNs with label noise is a very active research topic. Thus, many studies are being conducted "in parallel" along this direction. Thus, it seems possible that well-established techniques (e.g., early stopping) could be shared in these parallel studies. Nevertheless, the above-mentioned "merger" of early stopping and learning from the maximal safe set is our unique contribution. We have clarified this contribution to our updated paper (See Section 2 for details).

2.	Evaluation on more datasets such as Tiny-ImageNet and Clothing1M (Reviewer 2, 3)

We have conducted additional experiments for Tiny-ImageNet and Clothing1M. Prestopping maintained its dominance also in both datasets. In Tiny-ImageNet, Prestopping outperforms the existing methods by 1.7—13.9pp. In Clothing1M, Prestopping outperforms the existing methods by 0.7—5.8pp. Please see Appendix B.2 and B.3 of the updated paper.

3.	Comparison with more baselines (Reviewer 2, 3)

We could not add more baselines during the rebuttal period. However, many baselines (e.g., Iterative[4], S-model[9], F-correction[10]) have been reported to perform poorly than Co-teaching[6] we included, and we confirm that Prestopping outperforms more recent baselines according to the accuracy reported in the literature. For example, for CIFAR-10 and 30%-pair noise, the test accuracy of Prestopping was 90.8%, while the test accuracy of ITLM[8] was 88.2%; For CIFAR-10 and 40%-symmetric noise, the test accuracy of Prestopping was 88.7%, while the test accuracy of D2L[5] was 83.4%. We will definitely add more baselines to the camera-ready version, if accepted.

[9] Training Deep Neural Networks using A Noise Adaptation Layer (ICLR’17)
[10] Making Deep Neural Networks Robust to Label Noise: A Loss Correction Approach (CVPR’17)

4.	Comparison of sample selection using the small-loss trick and the maximal safe set (Reviewer 2)

We have added the performance evaluation of selecting clean samples using the small-loss trick (Co-teaching) and the maximal safe set (Prestopping). The selection accuracy of Prestopping was much higher (by up to 18.2pp) than that of Co-teaching (See Section 5.3 for details).

5.	Fair comparison: did the baselines use the clean validation set? (Reviewer 2)

For fair comparison, all the baselines used the clean validation sets to select the best number of epochs based on the validation accuracy. The best test error reported in our paper was evaluated using the network at the time of the lowest validation error. We have clarified this issue in the updated paper. (See Section 5 for details).

---

### Decision · Program_Chairs · 2019-12-19

**Decision:**

Reject

**Comment:**

This paper focuses on avoiding overfitting in the presence of noisy labels. The authors develop a two phase method called pre-stopping based on a combination of early stopping and a maximal safe set. The reviewers raised some concern about illustrating maximal safe set for all data sets and suggest comparisons with more baselines. The reviewers also indicated that the paper is missing key relevant publications. In the response the authors have done a rather through job of addressing the reviewers comments. I thank them for this. However, given the limited time some of the reviewers comments regarding adding new baselines could not be addressed. As a result I can not recommend acceptance because I think this is key to making a proper assessment. That said, I think this is an interesting with good potential if it can outperform other baselines and would recommend that the authors revise and resubmit in a future venue.